# SEMIALGEBRAIC NEURAL NETWORKS:
# FROM ROOTS TO REPRESENTATIONS

**S. David Mis**
Rice University
dmis@rice.edu

**Matti Lassas**
University of Helsinki
matti.lassas@helsinki.fi

**Maarten V. de Hoop**
Rice University
mdehoop@rice.edu

## ABSTRACT

Many numerical algorithms in scientific computing—particularly in areas like numerical linear algebra, PDE simulation, and inverse problems—produce outputs that can be represented by semialgebraic functions; that is, the graph of the computed function can be described by finitely many polynomial equalities and inequalities. In this work, we introduce Semialgebraic Neural Networks (SANNs), a neural network architecture capable of representing any bounded semialgebraic function, and computing such functions up to the accuracy of a numerical ODE solver chosen by the programmer. Conceptually, we encode the graph of the learned function as the kernel of a piecewise polynomial selected from a class of functions whose roots can be evaluated using a particular homotopy continuation method. We show by construction that the SANN architecture is able to execute this continuation method, thus evaluating the learned semialgebraic function. Furthermore, the architecture can exactly represent even discontinuous semialgebraic functions by executing a continuation method on each connected component of the target function. Lastly, we provide example applications of these networks and show they can be trained with traditional deep-learning techniques.

## 1 INTRODUCTION

Many classical numerical algorithms compute semialgebraic functions—functions whose graphs are defined by finitely many polynomial equalities and inequalities. Such functions include anything computable using finitely many real variables and finitely many operations addition, subtraction, multiplication, division, extraction of roots, and branching `if` statements with polynomial decision boundaries. Familiar primitives from numerical linear algebra, such as solving linear systems, Schur complements, and extraction of real eigenvalues are also semialgebraic, as are the solution operators of optimization problems with polynomial objective functions and constraints (Press et al., 2007). Polynomial functions of $n$-th roots $\sqrt[n]{\cdot}$ are widely used in computation of eigenvalues of non-symmetric matrices and in the perturbation theory of eigenvalues of linear operators (Kato, 1995). Also, Finite Element approximations of partial differential equations lead to solving linear systems whose solutions are semialgebraic functions (Beltzer, 1990). Due to their ubiquity, it is likely that the space of semialgebraic functions provides good inductive bias for operator learning with neural networks. This ansatz leads to our central question: Is it possible to design a neural network architecture capable of exactly representing any semialgebraic function?

In this paper, we combine techniques from classical numerical analysis, namely polynomial *homotopy continuation methods* for root finding, with ReLU activation functions to create neural networks capable of exactly representing any bounded semialgebraic function. The starting point for our idea is the observation that, under mild conditions, the graph of a semialgebraic function $F$ given by the relation $F(x) = y$ can be encoded as the kernel of a continuous piecewise polynomial $G$, i.e., the set of $(x, y)$ where $G(x, y) = 0$ (Section 2). Therefore, in principle, we can use a neural network combining polynomials and ReLU activations to learn $G$, then append a root-finding procedure such as Newton's method to compute $F(x) = \texttt{root}(G(x, \cdot)) = y$ in a manner similar to Bai et al. (2019). However, this straightforward approach has numerous practical difficulties: there is no guarantee such a neural network would have a root $y$ for every $x$, roots may not be unique, the root-finding procedure may never converge, etc. To remedy these issues, we take inspiration from homotopy continuation methods for root finding. We construct a function $H(x, y, s)$ that continuously deforms

from a simple function $G_0$ to the target function $G$:

$$H(x, y, 0) = G_0(x, y) \qquad H(x, y, 1) = G(x, y). \tag{1}$$

We use $H$ to find a root of $G$ by starting at a root of the simpler function $G_0$ at $s = 0$, then slowly increasing $s$ and tracking the motion of the root. In classical homotopy continuation methods, this procedure is carried out by numerically solving an ODE initial value problem describing the motion of $y$ as a function of $s$. With a properly constructed $H$, this procedure is powerful enough to compute the roots of $G$, even if $G_0$ bears little resemblance to $G$ (Theorem 37).

Instead of using a neural network to compute $G$ directly, our Semialgebraic Neural Networks (SANNs) compute the vector field of an ODE system arising from a homotopy continuation method to find a root of $G$ (Section 3). We then integrate across the interval $s \in [0, 1]$ using an off-the-shelf ODE solver. This approach is powerful enough to represent all bounded semialgebraic functions in the sense that $F(x)$ is the exact solution to the ODE initial value problem (Section 4). This approach always has a well-defined output, avoids costly explicit root-finding procedures, and has fixed evaluation time when using a non-adaptive ODE solver. To our knowledge, we present the first neural networks capable of computing arbitrary bounded semialgebraic functions on high-dimensional data.

## 1.1 RELATED WORK

**Implicit deep learning.** SANNs fall within the general framework of "implicit deep learning" (El Ghaoui et al., 2021; Kolter et al., 2020), particularly as popularized by Neural ODEs (Chen et al., 2019) and Deep Equilibrium Models (Bai et al., 2019). In particular, SANNs can be trained using the adjoint sensitivity method (Pontryagin et al., 1962) in a manner similar to Neural ODEs. Unlike Neural ODEs, where the ODE is fixed (based on network weights) and the initial value is based on the network input $x$, SANNs instead define a family of ODEs parameterized by $x$ and the initial value is fixed. Importantly, Neural ODEs compute diffeomorphisms (Dupont et al., 2019), while SANNs compute bounded semialgebraic functions, which may not be continuous or differentiable everywhere. Compared to Deep Equilibrium Models, the SANN architecture does not include an explicit root-finding or fixed-point computation step; instead SANNs directly parameterize a homotopy continuation method for root-finding.

**Semialgebraic machine learning.** There are many neural network architectures designed to represent important subsets of semialgebraic functions. For example, polynomials and piecewise polynomials are computed by de Hoop et al. (2022); Chrysos et al. (2021); Oh et al. (2003), and certain rational functions are computed by Boullé et al. (2020). We are unaware of prior neural networks designed to represent the entire class of bounded semialgebraic functions as a whole. An exciting complementary approach to semialgebraic approximation has been developed by Marx et al. (2021), where they encode semialgebraic functions as the optima of functions constructed from Christoffel–Darboux polynomials. Their approach is developed within the framework of classical approximation theory and data analysis rather than neural networks.

## 1.2 OUR CONTRIBUTION

We introduce the SANN architecture and provide an exact characterization of the functions it represents. Our key contributions are as follows:

**Definition of SANNs:** We present the SANN architecture (Algorithm 1), a new object grounded in classical homotopy continuation methods for root-finding, that is capable of exactly representing any bounded semialgebraic function.

**Exact characterization of expressivity:** We rigorously characterize the expressivity of SANNs (Section 4), proving their ability to represent all continuous bounded semialgebraic functions via a homotopy continuation argument (Section 4.1).

**Extension to discontinuous functions:** Building on the homotopy continuation framework, we extend our results to show that SANNs can also represent discontinuous bounded semialgebraic functions through constructive proofs (Section 4.2).

We also argue SANNs are naturally suited to a wide variety of difficult problems, with applications to solving linear systems (Section 5), nonlinear inverse problems (Section F), general optimization

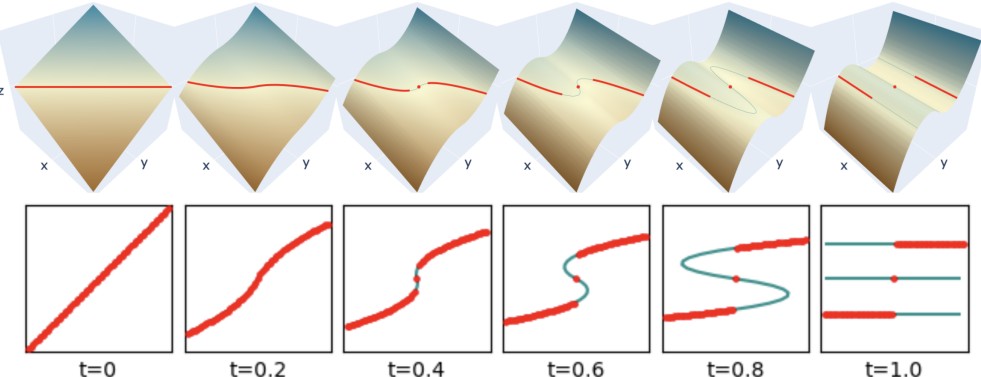

Figure 1: Computing the Heaviside step function $F$ (plotted in red) using a homotopy method. The graph of $F$ is a subset of the kernel of a higher-dimensional piecewise polynomial $G$. In the upper row, we plot the surface of a homotopy $H$ such that $H(x, y, 0) = 0$ is trivial to solve (upper left), and $H(x, y, 1) = G$ (upper right). The kernel of $H$ is shown with a green line (red points are also on top of the green line). The bottom row shows the projection of $H$ onto to $xy$-plane. We compute $F(x) = y$ by following the kernel of $H$ from time $t = 0$ to $t = 1$, keeping $x$ fixed. Although the kernel of $G$ contains points outside the graph of $F$ (the visible parts of the green lines), these points are never encountered when computing $F$ by following the homotopy. This process captures the discontinuity exactly, including the isolated point at $(0, 0)$.

problems (Section G.1), and transformers (Section G.3). We also demonstrate through numerical experiments that SANNs can be trained using standard techniques (Sections 5.2 and F.1), and discuss future research directions (Section 6).

## 2 BACKGROUND

### 2.1 INF-SUP DEFINABLE PIECEWISE POLYNOMIALS

Our networks are constructed from the piecewise polynomials formed by combining polynomials with ReLU activations. Using these components, we can represent the pointwise minimum and maximum of any finite collection of polynomials. In mathematical literature, such structures are referred to as *lattices*. Instead of "min" and "max," the terms "inf" and "sup" are traditionally used in lattice theory; since our constructions rely heavily on lattice theory, we adopt this terminology here. Notably, we do not use the word "lattice" to refer to a grid-like structure (e.g., $\mathbb{Z}^d$), but rather a set of functions that is closed under the "min" and "max" operations.

We begin by defining the class of functions that form the backbone of our constructions. The notation $\mathbb{R}[x_1, \ldots, x_m]$ refers to the set of polynomials of variables $x_1, \ldots, x_m$ with coefficients from $\mathbb{R}$.

**Definition 1** (Inf-sup definable piecewise polynomials). Let $D \subseteq \mathbb{R}^m$. $ISD(D)$ is the lattice generated by the polynomials $\mathbb{R}[x_1, \ldots, x_m]$, viewed as functions $D \to \mathbb{R}$, together with the min and max operations. The vectors of $n$ $ISD(D)$ functions are denoted $ISD(D, \mathbb{R}^n)$. For any non-negative integer $k$, $ISD^k(D, \mathbb{R}^n)$ are the $k$-times differentiable functions in $ISD(D, \mathbb{R}^n)$.

The shorthand "$f$ is $ISD$" means $f \in ISD(D, \mathbb{R}^n)$ for some appropriate $D$ and $n$. Our nomenclature is taken from Mahé (1984) and Mahé (2007).

Summarizing, the components $f_k$ of a function $f \in ISD(D, \mathbb{R}^n)$, $D \subset \mathbb{R}^m$ can be written as

$$f_k(x) = \max_{i=1,\ldots,I} \min_{j=1,\ldots,J} \left( \sum_{\alpha_1, \alpha_2, \ldots, \alpha_m = 0}^{N} a_{k,i,j,\alpha_1,\ldots,\alpha_m} x_1^{\alpha_1} x_2^{\alpha_2} \ldots x_m^{\alpha_m} \right), \quad x = (x_1, x_2, \ldots, x_m).$$

Later, when we consider $f_k$ as a layer of a neural network, the coefficients $a_{k,i,j,\alpha_1,\ldots,\alpha_m}$ are used as the parameters which are optimized in training process.

Since the min and max operations can be written using the ReLU function, all ISD functions can be written as a feed forward neural network where in the first hidden layer one has several different polynomial activation functions followed by fully connected ReLU layers. Furthermore, Henriksen & Isbell (1962) showed $ISD(D, \mathbb{R}^k)$ forms a ring, so the sum and product of any ISD function is itself ISD (see Theorem 22 in Appendix A). Thus $ISD(D, \mathbb{R}^k)$ are precisely the functions $D \to \mathbb{R}^k$ that can be represented using finitely many, arbitrarily interleaved vector additions, Hadamard products, scalar multiplications and ReLU activations on variables from $D$.

We focus our attention on ISD functions since these are the piecewise polynomials computable using neural networks that combine polynomials and ReLU activations. There are many ways to construct such networks; we provide an example based on *Operator Recurrent Neural Networks* (de Hoop et al., 2022) in Appendix B, along with proofs of the expressivity of that architecture. Our approach can be adapted to analyze the expressivity of other polynomial networks with ReLU activations as well, such as those proposed by Chrysos et al. (2021).

It is currently unknown whether every continuous real piecewise polynomial is ISD; this is the famous *Pierce–Birkhoff conjecture* (Bochnak & Efroymson, 1980; Madden, 2011). ISD functions suffice for our purpose to build semialgebraic neural networks, regardless whether the Pierce–Birkhoff conjecture ultimately holds or not.

**Definition 2** (Locally ISD functions). A function $f : \mathbb{R}^m \to \mathbb{R}^n$ is *locally ISD*, denoted $f \in ISD_{loc}(\mathbb{R}^m, \mathbb{R}^n)$, if for every bounded $D \subset \mathbb{R}^m$, $f|_D \in ISD(D, \mathbb{R}^n)$.

Appendix A contains additional background information on lattice theory.

## 2.2 SEMIALGEBRAIC GEOMETRY

We now present the basic properties of semialgebraic sets relevant to SANNs. A more thorough introduction to semialgebraic geometry can be found in the first chapters of Bochnak et al. (1998).

**Definition 3** (Basic semialgebraic set). A set $D \subset \mathbb{R}^m$ is a *basic semialgebraic set* if there exists $J_1, J_2 \in \mathbb{N}$ and finite sets of polynomials $P = \{p_i\}_{i=1}^{J_1}$, $Q = \{q_i\}_{i=1}^{J_2} \subset \mathbb{R}[x_1, \ldots, x_m]$ such that
$$D = \{x \in \mathbb{R}^m \mid p(x) = 0,\ q(x) < 0 \text{ for all } p \in P,\ q \in Q\}.$$

**Definition 4** (Semialgebraic set). A set $D \subset \mathbb{R}^m$ is a *semialgebraic set* if it is the union of finitely many basic semialgebraic sets.

Finite unions, finite intersections, complements, projections, closures, and interiors of semialgebraic sets are all semialgebraic themselves, a result of the famous Tarski–Seidenberg theorem (Tarski, 1951; Seidenberg, 1954) (on the formulation of the Tarski–Seidenberg theorem we use in this paper, see Appendix E.9). Furthermore, every closed semialgebraic set can be defined using only non-strict inequalities (Lojasiewicz, 1964).

**Definition 5** (Semialgebraic function). $f : \mathbb{R}^m \to \mathbb{R}^n$ is a *semialgebraic function* if its graph is a semialgebraic subset of $\mathbb{R}^m \times \mathbb{R}^n$.

Special cases of semialgebraic functions include (piecewise) polynomials and (piecewise) rational functions, among others. For example, the graph of $x \mapsto 1/x$ is the set $\{(x, y) \in \mathbb{R}^2 \mid xy = 1\}$.

**Example 6.** Consider the semialgebraic function $F : \mathbb{R} \to \mathbb{R}$ below.

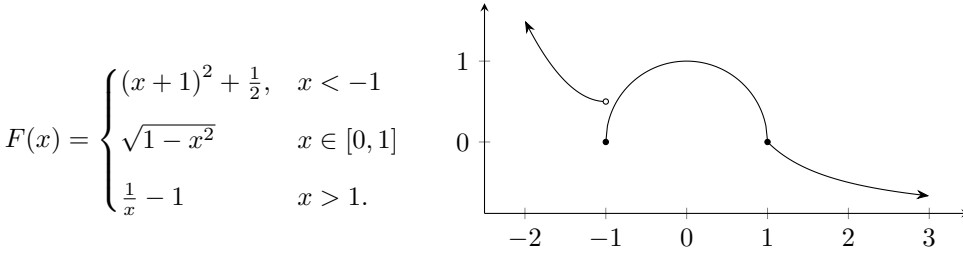

$$F(x) = \begin{cases} (x+1)^2 + \frac{1}{2}, & x < -1 \\ \sqrt{1 - x^2} & x \in [0, 1] \\ \frac{1}{x} - 1 & x > 1. \end{cases}$$

$F$ is a polynomial when $x < 1$, a rational function when $x > 1$, and neither when $x \in [0, 1]$. It has a single discontinuity at $x = -1$, and its graph is not closed there. It is unbounded as $x \to -\infty$, but becomes bounded whenever the domain is restricted from below.

## 2.3 SEMIALGEBRAIC FUNCTIONS AS KERNELS OF ISD FUNCTIONS

In this subsection, we show how to represent semialgebraic sets as the kernel of ISD functions. This is a semialgebraic analogue to the standard technique in differential geometry of constructing smooth manifolds from the level sets of smooth functions (Lee, 2013). Proofs for Proposition 7 and Corollary 8 as well as additional commentary and an illustration can be found in Appendix C.

**Proposition 7.** $S \subset \mathbb{R}^m$ *is a closed semialgebraic set if and only if there exists* $f \in ISD^1(\mathbb{R}^m, \mathbb{R}_{\geq 0})$ *such that* $\ker(f) = S$, *where* $\ker(f) := \{x \in \mathbb{R}^m : f(x) = 0\}$.

We extend the above theorem to graphs of vector-valued semialgebraic functions with closed graphs.

**Corollary 8.** $F : \mathbb{R}^m \to \mathbb{R}^n$ *is a semialgebraic function with closed graph if and only if there exists a* $G \in ISD^1(\mathbb{R}^m \times \mathbb{R}^n, \mathbb{R}^n_{\geq 0})$ *such that* $\ker(G) = gr(F)$, *where* $gr(F) := \{(x, F(x))\} \subset \mathbb{R}^m \times \mathbb{R}^n$ *is the graph of* $F$.

Corollary 8 makes it computationally possible to find points of the graph of the semialgebraic function $F$ by using a SANN related to an ISD function $G$, as we will see in Section 4.

## 3 SEMIALGEBRAIC NEURAL NETWORKS

### 3.1 ISD NETWORKS

SANNs are built from auxiliary networks capable of computing ISD functions; we call such networks "ISD networks". As discussed in Section 2, there are various ways to design such networks. An example architecture and its expressivity theorems are provided in Appendix B.

To streamline notation, we define a class of ISD functions that will be used frequently:

**Definition 9.** $ISDnet(m, n, k) := ISD(\mathbb{R}^m \times \mathbb{R}^{n+k} \times \mathbb{R}, \mathbb{R}^{(n+k) \times (n+k)} \times \mathbb{R}^{n+k})$

A function in $ISDnet(m, n, k)$ accepts three parameters $x \in \mathbb{R}^m$, $z \in \mathbb{R}^{n+k}$, and $s \in \mathbb{R}$, and it returns a matrix $M \in \mathbb{R}^{(n+k) \times (n+k)}$ and vector $b \in \mathbb{R}^{(n+k)}$. We further decompose $z = (y, t)$, where $y \in \mathbb{R}^n$ is the output of the computed semialgebraic function and $t \in \mathbb{R}^k$ are auxiliary "time" variables needed for the homotopy continuation constructions in Section 4. Including these auxiliary variables is analogous to "lifting" (i.e. increasing the width) common in many architectures. Such lifting is required for narrow ReLU networks to be universal approximators in the infinite-layer limit (Lu et al., 2017). A similar strategy is used in Dupont et al. (2019) to augment Neural ODEs.

We require only $k = 1$ in our constructions, but it is easy to extend the arguments to $k > 1$. Larger values of $k$ may have practical benefits not captured by the characterization of the range of SANNs presented here, and we leave that investigation for future work.

### 3.2 SANN ARCHITECTURE

We define SANNs to be functions of the form $f_{\mathcal{N}, c_{\max}} : x \mapsto \Pi z_N$, where $\mathcal{N} \in ISDnet(m, n, k)$, $N \in \mathbb{N}$, $c_{\max} > 0$, $x \in \mathbb{R}^m$, and $z_N$ is obtained by an approximation of the solution of an ordinary differential equation, that is, by setting $z_0 = 0$ and defining for $j = 0, 1, \ldots, N - 1$,

$$z_{j+1} = \text{ODE-step}\left(\dot{z}_j, z_j, \frac{j}{N}\right) := z_j + \frac{1}{N}\dot{z}_j \tag{2}$$

where

$$\dot{z}_j = \text{clamp-sol}\left(M\left(x, z_j, \frac{j}{N}\right), b\left(x, z_j, \frac{j}{N}\right)\right). \tag{3}$$

Matrix $M(x, z, s)$ and vector $b(x, z, s)$ are the output of an ISD network $\mathcal{N}(x, z, s)$. The function clamp-sol is defined

$$\text{clamp-sol}(M, b) := \begin{cases} \text{clamp}(M^{-1}b, -c_{\max}, c_{\max}) & \text{if } M \text{ is invertible} \\ 0 & \text{otherwise.} \end{cases} \tag{4}$$

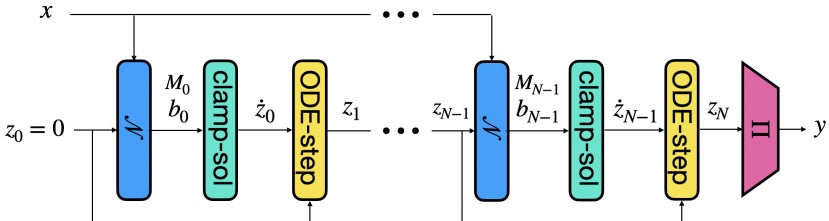

Figure 2: Architecture diagram for a SANN. The SANN outputs $y$ as a semialgebraic function of input $x$. Vectors $z_j$ are the current values of an ODE at timestep $j$. The time-derivative $\dot{z}_j$ is computed using a neural network $\mathcal{N}$ capable of computing ISD piecewise polynomials. $\mathcal{N}$ accepts the current ODE state $(z_j, j/N)$, as well as recurrent input $x$. The output of $\mathcal{N}$ is a matrix $M$ and vector $b$, from which $\dot{z}_j$ is computed using clamp-sol$(M, b)$ ($\dot{z}_i = M^{-1}b$ in the common case). ODE-step is a single update of a numerical ODE solver. Finally, $\Pi$ is a projection operation such that $y$ is the first $n$ components of $z$.

The function "clamp" operates component-wise on each element of a vector, and is defined

$$\text{clamp}(a, low, high) := \begin{cases} a, & a \in [low, high] \\ low, & a < low \\ high & a > high. \end{cases} \tag{5}$$

These functions guarantee the iteration (3) is well-defined even when $\mathcal{N}$ produces singular or nearly-singular output matrices $M$. Lastly, $\Pi$ projects onto the first $n$ components of its input.

The iteration (3) is the Euler finite-difference scheme that approximates the solution of the ODE

$$\dot{z}(s) = \text{clamp-sol}(M(x, z(s), s), b(x, z(s), s)), \tag{6}$$
$$z(0) = 0. \tag{7}$$

As $N \to \infty$, the output of the finite difference scheme converges to the solution of the ODE. These limit functions are of the form $f_{\mathcal{N}, c_{\max}}^{lim} : x \mapsto \Pi z(1)$ that are defined by the ODE (6)–(7). We call these functions as the limit functions of SANNs and denote those by $SANN^{lim}(\mathcal{N}, \cdot, c_{\max})$. We note that the forward-Euler definition for ODE-step used in (2) can be replaced with other numerical solvers, such as Runge-Kutta methods.

### 3.3 SANN ARCHITECTURE: ALGORITHMIC PRESENTATION

The above definition of the SANN architecture is amenable to mathematical analysis. Algorithm 1 is an alternative presentation that closely reflects how a SANN may be implemented in a functional programming style. It is completely equivalent to the description above.

---
**Algorithm 1** Evaluating a SANN

---
**Require:** $\mathcal{N} \in ISDnet(m, n, k)$
$\qquad x \in \mathbb{R}^m$
$\qquad c_{\max} \in \mathbb{R}_{\geq 0}$
1: **function** SANN($\mathcal{N}, x, c_{\max}$)
2: $\quad$ **function** $\dot{z}(z, s)$
3: $\qquad$ $(M, b) \leftarrow \mathcal{N}(x, z, s)$
4: $\qquad$ **if** $M$ is singular **then**
5: $\qquad\quad$ **return** 0
6: $\qquad$ **else**
7: $\qquad\quad$ **return** clamp($M^{-1}b, -c_{\max}, c_{\max}$)
8: $\quad$ $(y, t) \leftarrow$ ODESolve($\dot{z}, (0, 0)$)
9: $\quad$ **return** $y$

---

In Algorithm 1, `ODESolve`$(\dot{z}, z(0))$ outputs $z_N$ according to the finite difference scheme in formulas (2), or another numerical solver chosen by the programmer. This interface reflects popular libraries for numerically solving ODEs, e.g. Malengier et al. (2018); Kidger (2021).

*Remark.* Our expressivity proofs below refer to the true solution $z(1)$ to the ODE (6)–(7) rather than the numerical solution $z_N$. The numerical ODE solver may introduce approximation error into SANNs, even for semialgebraic functions, but this approximation error is of a fundamentally different character than that which normally arises in neural networks, and the mitigation strategies will be different for SANNs. For example, increasing the number of steps $N$ will decrease the approximation error *without* increasing the number of parameters in the network. On the other hand, increasing the number of parameters in $\mathcal{N}$ does not decrease the approximation error, as it generally would for other architectures. Furthermore, SANNs may be constructed using other numerical strategies besides the finite-difference scheme (2). Changing this scheme does not affect our expressivity proofs in this paper, but there may be other implications in practice.

## 3.4 TRAINING

In the supervised learning regime, we are given training data $\{(x^{(\ell)}, y^{(\ell)}) : \ell = 1, 2, \ldots, L\}$, $y^{(\ell)} = F(x^{(\ell)})$ where $F$ is the target semialgebraic function. The output of a SANN with input $x$ is $y_N(x) := \Pi z_N$ and the intermediate update directions given in equation (3) are $\dot{y}_j(x) := \Pi \dot{z}_j$. In our experiments, we found it helpful to include a "direction loss" term to train each $y_j$ for $j = 1, \ldots, N$:

$$\mathcal{L}_{\text{direction}}(x^{(\ell)}) := \frac{1}{N} \sum_{j=1}^{N} \left\| y^{(\ell)} - \left( y_j(x^{(\ell)}) + \left(1 - \frac{j}{N}\right) \dot{y}_j(x^{(\ell)}) \right) \right\|^2, \tag{8}$$

$$\mathcal{L}_{\text{accuracy}}(x^{(\ell)}) := \left\| y^{(\ell)} - y_N(x^{(\ell)}) \right\|^2, \qquad \mathcal{L}_{\text{total}} := \sum_{\ell=1}^{L} \mathcal{L}_{\text{accuracy}}\left(x^{(\ell)}\right) + \lambda \mathcal{L}_{\text{direction}}\left(x^{(\ell)}\right) \tag{9}$$

where $\lambda > 0$ is a small, suitably chosen parameter, e.g. $\lambda = 10^{-2}$. $\mathcal{L}_{\text{direction}}$ reflects the accuracy loss if the direction vector $\dot{y}_j$ were held constant for the remainder of the numerical integration.

## 4 EXPRESSIVITY OF SANNS

In this section, we show that SANNs are capable of exactly representing any bounded semialgebraic function. We start by showing that every continuous semialgebraic function can be evaluated by a particular homotopy continuation method that is computable by iteratively solving the ODE (6)–(7) with different parameter functions $\mathcal{N}$. We then extend this result to cover discontinuous semialgebraic functions by first separating each connected component of the graph, applying results for the continuous case to obtain homotopies that represent each piece, then glueing these homotopies together in a manner consistent with ODE (6)–(7) to obtain the SANN. Additional details, including background in semialgebraic geometry and full proofs of the theorems we present here, are in Appendices D and E.

### 4.1 CONTINUOUS SEMIALGEBRAIC FUNCTIONS

This section introduces a family of homotopies $H_{x,a}$ from which we can construct a continuation method to evaluate the roots of ISD functions.

Let $u_F < u_H$ be positive real numbers and $U_F = (-u_F, u_F)^n$. Consider the family of homotopies $H_{x,a} : \mathbb{R}^n \times \mathbb{R} \to \mathbb{R}^n$, parameterized by $(x, a) \in \mathbb{R}^m \times U_F$, defined

$$H_{x,a}(y, t) := (1 - t)(y - a) + t\big(g(x, y) + G(x, y)\big) \tag{10}$$

where $g \in ISD^1_{loc}(\mathbb{R}^m \times \mathbb{R}^n, \mathbb{R}^n_{\geq 0})$ and $G \in ISD^1(\mathbb{R}^m \times \mathbb{R}^n, \mathbb{R}^n)$. Each $H_{x,a}$ is constructed from ring operations on locally ISD functions, so it is locally ISD itself.

We always evaluate $H_{x,a}$ keeping $x$ fixed; to simplify notation, let $g_x := g(x, \cdot)$ and $G_x := G(x, \cdot)$.

For our construction to work, we require $g_x$ to be 0 on $U_F$ and grow quickly outside $U_F$. Specifically, for every $x \in \mathbb{R}^m$, we require $G_x = o(g_x)$ and $\text{id}_y = o(g_x)$, where we have used Landau "little-o" notation; e.g. $G_x = o(g_x)$ means $\lim_{\|y\| \to +\infty} \frac{G_x(y)}{g_x(y)} = 0$.

The next lemma shows how to construct $g_x$ based only on $U_F$.

**Lemma 10.** *There exists $g \in ISD^1_{loc}(\mathbb{R}^m \times \mathbb{R}^n, \mathbb{R}^n)$ such that $g_x \equiv 0$ on $U_F$ and for every $G \in ISD^1(\mathbb{R}^m \times \mathbb{R}^n, \mathbb{R}^n)$, we have $G_x = o(g_x)$.*

*Proof.* Let $u_0 = u_F$, $u_j = u_{j-1} + 1$, $U_j = (-u_j, u_j)^n$ for $j \in \mathbb{N}$. Notice $\mathbb{R}^n = \bigcup_{j=1}^\infty U_j$. Define $g$ piecewise as follows:

$$g(x,y) = \begin{cases} 0 & y \in U_F \\ \sum_{j=1}^J (\mathrm{ReLU}(\mathrm{sign}(y)y - u_{j-1}\mathbf{1}))^{j+1} & y \in U_J \setminus U_{J-1}, \text{ for each } J \in \mathbb{N}. \end{cases} \quad (11)$$

(The exponent $j + 1$ denotes repeated component-wise multiplication.) This $g$ is not ISD since it is defined over infinitely many regions, but it is locally ISD. Furthermore, the local degree of $g$ increases without bound as $\|y\| \to \infty$, so $g$ eventually dominates any polynomial. Thus $G_x = o(g_x)$ for any ISD $G$. Finally, it is straightforward to verify $g$ is continuously differentiable. $\square$

The next theorem shows the family of homotopies in equation (10) is capable of representing any continuous bounded semialgebraic function $F$ for almost every choice of $a$; more precisely every $a$ except possibly on a semialgebraic set of dimension less than $n$ (c.f. the Transversality theorem in D.1). The requirement that $F$ be bounded is always satisfied when the domain of continuous $F$ is compact, which is a common and reasonable assumption in the context of machine learning with finite training data. The proof of Theorem 11 is in appendix E.

**Theorem 11.** *Let $F : \mathbb{R}^m \to U_F$ be a given continuous semialgebraic function. For every $x \in \mathbb{R}^m$ there exists $H_{x,a}$ with the form (10) such that for almost every $a \in U_F$, the following hold:*

1. *There exists $y \in ISD([0,1], \mathbb{R}^n), t \in ISD([0,1], \mathbb{R})$, such that $t(0) = 0$, $t(1) = 1$, and $(y(s), t(s)) \in \ker(H_{x,a})$ for all $s \in [0,1]$.*

2. *The kernel of $H_{x,a}(\cdot, 1)$ is the singleton $\{F(x)\}$.*

Theorem 11 is used as follows: When $t = 0$ the function $y \mapsto H_{x,a}(y, 0)$ has the root $a$. We apply the curve-tracing algorithm by solving the ODE (30)–(33) (see Appendix D.3) for the function $H_{x,a}$ to find roots of the functions $H_{x,a}(\cdot, t)$ with $t \in [0, 1]$. The root of $H_{x,a}(\cdot, 1)$ gives $y = F(x)$.

Conclusion 1 guarantees the homotopy $H_{x,a}(y, t)$ can be solved to time $t = 1$ for almost every choice of $a$; SANNs correspond to the choice $a = 0$. Conclusion 2 guarantees this solution is indeed the desired value $F(x)$.

To prove that SANNs can represent any continuous semialgebraic function, we need only show that the ODE (6)–(7) defining SANNs can recapitulate the ODE (30)–(33) defining the $H_{x,a}$ homotopy continuation method. A full proof is in Appendix E.

**Theorem 12.** *Let $F : \mathbb{R}^m \to U_F$ be a given continuous semialgebraic function. Then there exists $\mathcal{N} \in ISDnet(m, n, 1)$ and $c_{\max} > 0$ such that $SANN^{lim}(\mathcal{N}, \cdot, c_{\max}) = F$.*

## 4.2 Discontinuous semialgebraic functions

Discontinuities can arise in the ODE (6)–(7) since the value of $\dot{z} = M^{-1}b$ can be discontinuous across a boundary where $M$ is singular. Consider the simple example

$$M = [\mathrm{abs}(x)] \qquad b = [x] \qquad \dot{z} = \text{clamp-sol}(M, b) = \begin{cases} -1 & x < 0 \\ 1 & x > 0, \end{cases} \quad (12)$$

where the clamp-sol operation is defined with $c_{max} = 2$. We exploit this behavior to show via construction that SANNs can exactly represent even discontinuous semialgebraic functions.

**Theorem 13.** *Let $F : \mathbb{R}^m \to \mathbb{R}^n$ be a (possibly discontinuous) bounded semialgebraic function. Then there exists $\mathcal{N} \in ISDnet(m, n, 1)$ and $c_{\max} \in \mathbb{R}_{\geq 0}$ such that $SANN^{lim}(\mathcal{N}, \cdot, c_{\max}) = F$.*

Every semialgebraic set has finitely many semialgebraic connected components, so every semialgebraic function is piecewise continuous on finitely many pieces. Our approach is to first show how we are able to represent the characteristic function of semialgebraic sets using the ODE (6)–(7). From these, we can represent a semialgebraic decomposition of the domain $\mathbb{R}^m$, and apply the continuous homotopy arguments above on each continuous region. We finally glue everything together in a way consistent with solving the ODE (6)–(7). The full proof is in Appendix E.

## 5 NUMERICAL EXAMPLE: SOLVING LINEAR SYSTEMS

In this section, we give an example of a numerical algorithm that can be exactly represented by a SANN. Specifically, we construct a SANN that exactly computes (to machine precision) the solution to a linear system $Xy = b$, something that is not possible for standard neural network architectures. The parameters for this exact reconstruction are chosen by hand; we also show that SANNs can be trained from data using traditional techniques to perform comparably to feed-forward networks.

We focus on the Jacobi iteration method for solving dense linear systems $Xy = g$. Split $X$ into its diagonal part $D$, upper-triangular part $U$, and lower-triangular part $L$.

$$X = \begin{bmatrix} d_1 & & U \\ & \ddots & \\ L & & d_n \end{bmatrix}, \quad D = \text{diag}(d_1, d_2, \ldots, d_n). \tag{13}$$

From any initial guess $y_0$, the Jacobi iteration method computes the iteration $y_j$ using

$$y_{j+1} = D^{-1}\big(g - (L+U)y_j\big). \tag{14}$$

Adding $y_j - y_j$ to the right side and simplifying yields

$$y_{j+1} = y_j + D^{-1}\big(g - Xy_j\big). \tag{15}$$

The sequence of iterates $(y_1, y_2, \ldots)$ computed in this way converges to the exact solution $y^* = X^{-1}b$ when the spectral radius of $D^{-1}(L+U)$ is less than 1. It is exactly of the form (2), and thus computable by a SANN with an appropriate ISD network $\mathcal{N}$.

An additional numerical example addressing a nonlinear inverse problem for electrical resistor networks is in Appendix F. The problem is to find the values of an unknown resistivity function on a graph from the voltage-to-current measurements on a subset of the nodes of the graph. This problem is encountered in Electrical Impedance Tomography (Borcea, 2002), a medical imaging modality, and its mathematical treatment is based on a Finite Element approximation of a partial differential equation (Lassas et al., 2015).

### 5.1 EXACT INVERSION USING A HAND-CRAFTED NETWORK

To demonstrate the new possibilities afforded by the expressive power of SANNs, we manually configured the parameters of a SANN to replicate the classical Jacobi iteration for solving linear systems. With this setup, the SANN is able to solve the system to machine precision—something standard feed-forward networks, which compute only piecewise-linear functions, cannot achieve. Figure 3 shows the error of $y_j = \Pi z_j$ decrease to machine precision for several inputs $(X, g)$

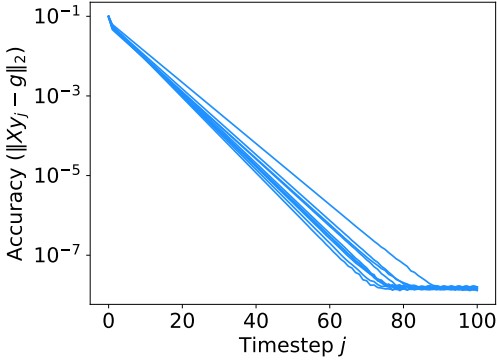

Figure 3: Trajectories of a SANN that solves linear systems to machine precision. Given a matrix $X \in \mathbb{R}^{50 \times 50}$ and vector $b \in \mathbb{R}^{50}$, the network produces a vector $y \in \mathbb{R}^{50}$ such that $Xy = b$. Each line shows the trajectory of the output $y$ for a given input pair $(X, g)$ as the SANN's ODE is iteratively solved. At each timestep, the network performs a single Jacobi iteration update. The output of the network exactly matches the output of 100 steps of Jacobi iteration.

### 5.2 RESULTS OF TRAINING

To demonstrate the feasibility of training SANNs, we trained a SANN to solve $10 \times 10$ linear systems; given matrix $X \in \mathbb{R}^{10 \times 10}$ and vector $g \in \mathbb{R}^{10}$, find vector $y \in \mathbb{R}^{10}$ such that $Xy = g$.

The SANN uses a 2-layer Matrix-Recurrent Neural Network (see appendix B) as the underlying ISD network $\mathcal{N}$. We used Adam optimizer to minimize the loss $\mathcal{L}_{\text{total}}$ from equation (8).

To benchmark SANNs against standard techniques, we trained a two-layer feed-forward neural network with a comparable number of parameters on the same task. Table 1 summarizes the results. In this experiment, the SANN achieved marginally better accuracy with slightly fewer parameters.

|  | # parameters | Validation accuracy $\|Xy - g\|_2$ |
| --- | --- | --- |
| Feed forward network | $306, 140$ | 0.166 |
| SANN | $\mathbf{291, 720}$ | **0.101** |

Table 1: Results of training a SANN and feed-forward neural network to solve linear systems. Given a $10 \times 10$ matrix $X$ and and vector $g$, the networks output $y$ such that $Xy \approx g$.

While we refrain from claiming significant numerical advantages for SANNs at this stage, the results underscore the feasibility of training these architectures.

The input matrices $X$ were sampled from a distribution of strictly diagonally dominant matrices, ensuring that a SANN theoretically exists that can solve for $y$ to machine precision (as in the hand-crafted example from the previous subsection). However, further research is needed to determine how to effectively optimize network parameters to achieve such precision.

## 6 Discussion and future work

We consider the SANN architecture presented in this paper to be the most general form of a new family of neural networks designed to compute semialgebraic functions. In practice, this architecture can be restricted in various ways to improve performance on specific tasks. This is analogous to convolutional neural networks (CNNs), which are a subset of standard feed-forward neural networks. While CNNs have far fewer parameters and cannot compute arbitrary piecewise-linear functions, they excel at tasks involving translational symmetry (e.g., image classification). Similarly, in future work, we aim to identify and develop specialized variants of the SANN architecture that compute subsets of semialgebraic functions tailored to particular tasks.

One interesting modification is to change line 3 of Algorithm 1 to have the ISD network $\mathcal{N}$ output an $LU$ factorization of $M$ rather than $M$ itself. This change would increase the speed of the networks since solving two triangular systems $LUz = b$ is significantly faster than solving a general linear system $Mz = b$. Furthermore, we could create a continuous variant of SANNs by requiring the diagonal elements of $L$ and $U$ to be positive and bounded away from 0. Such changes likely have many practical advantages; however, our expressivity proofs would no longer directly apply. Not every ISD matrix $M$ has an $LU$ decomposition in terms of ISD $L$ and $U$, and it is not clear whether this would be sufficient to apply our homotopy continuation arguments. Regardless, these modified SANNs compute a large class of semialgebraic functions not possible for most other architectures.

Efficient training of SANNs remains an open challenge. It is currently feasible to train SANNs using established techniques such as backpropagation when explicit time-stepping is used to evaluate the network's ODE (as we have done here). The adjoint sensitivity method (Pontryagin et al., 1962) can be used when other ODE solvers are employed, as in Neural ODEs (Chen et al., 2019). However, the unique structure of the SANN architecture offers opportunities for novel training strategies. New approaches are needed to take full advantage of the expressive power of SANNs.

## 7 Conclusion

We have presented new representation theorems for semialgebraic functions along with a novel neural network architecture capable of representing all bounded semialgebraic functions. Our methods are inspired by homotopy continuation methods for root finding, and the architecture is simple to implement using existing machine learning tools. We believe our SANNs build a new bridge between semialgebraic geometry (also called "real algebraic geometry") and machine learning, opening new avenues for both theoretical exploration and practical applications.

ACKNOWLEDGMENTS

The authors thank the anonymous ICLR reviewers for helpful comments on an ealier version of this manuscript. M. V. de Hoop and S. D. Mis acknowledge the support of the Simons Foundation under the MATH + X Program, the Department of Energy under grant DE-SC0020345, the Occidental Petroleum Corporation and the corporate members of the Geo-Mathematical Imaging Group at Rice University. M. Lassas was partially supported by a AdG project 101097198 of the European Research Council, Centre of Excellence of Research Council of Finland and the FAME flagship of the Research Council of Finland (grant 359186). Views and opinions expressed are those of the authors only and do not necessarily reflect those of the funding agencies or the EU. Neither the European Union nor the granting authority can be held responsible for them.

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

## A  LATTICE-ORDERED RINGS OF PIECEWISE POLYNOMIALS

This appendix introduces the classical theory of "lattice-ordered" and "function" rings, culminating in a statement of Henricksen's and Isbell's theorem for generating $f$-rings from subrings. This is the key lemma in our proof that the space of functions computable by MRNNs is closed under ring and lattice operations—vector addition, Hadamard product, min and max.

We use the following standard conventions: Group operations are denoted "$+$", and ring multiplication operations are denoted "$*$". When there is no ambiguity, the ring multiplication operation

$f * g$ may be written $fg$. If $R$ is a given ring, the polynomial ring in $n$ variables with coefficients in $R$ is denoted $R[X_1, \ldots, X_n]$. Direct products are denoted "$\times$", and we reserve $\otimes$ for Kronecker products of matrices. Lattice operations are denoted $\wedge$ for "inf" ("meet") and $\vee$ for "sup" ("join"). We use the shorthand $f^+ := f \vee 0$, $f^- := (-f) \vee 0$, and $|f| := f \vee (-f)$. In this paper,

$$f \vee g = \max(f, g), \quad \text{and} \quad f \wedge g = \min(f, g),$$

although in general there are other choices for the lattice operations. Observe also that

$$f \vee g = f + \text{ReLU}(g - f), \quad \text{and} \quad f \wedge g = f - \text{ReLU}(f - g).$$

Furthermore, $f \leq g$ means $f \wedge g = f$ and $f \vee g = g$, while the obvious dual statement holds for $f \geq g$.

In the terminology of neural networks, $f^+$ is precisely $\text{ReLU}(f)$ when the lattice is built over $\mathbb{R}^n$ and the meet/join operations are component-wise min/max.

**Definition 14** ($\ell$-rings). An $\ell$-*ring* (short for "lattice-ordered ring") is a tuple $(R, +, *, \wedge, \vee)$ that is a ring with operations $+, *$, a lattice with operations $\wedge, \vee$, and the following compatibility conditions are satisfied:

$$f \leq g \Rightarrow h + f \leq h + g \text{ for all } f, g, h \in R, \tag{16}$$

$$f \geq 0 \text{ and } g \geq 0 \quad \Rightarrow \quad fg \geq 0. \tag{17}$$

We will often refer to an $\ell$-ring $(R, +, *, \wedge, \vee)$ simply by its underlying set $R$.

An $\ell$-ring homomorphism preserves both ring an lattice operations.

*Remark.* The original masters did not require $\ell$-rings to have commutative addition (i.e. be an Abelian groups) or have multiplicative identities. These generalizations are not important to us here, and what we call "$\ell$-ring" is often called "$\ell$-ring with unity" in the literature. All our $\ell$-rings are abelian groups and have multiplicative identity.

Every $\ell$-ring is a distributive lattice[1] (Birkhoff, 1948); that is,

$$f \vee (g \wedge h) = (f \vee g) \wedge (f \vee h) \quad \text{and} \quad f \wedge (g \vee h) = (f \wedge g) \vee (f \wedge h)$$

hold for all $f, g, h \in R$. In particular, every lattice polynomial can be written in the form

$$\bigwedge_{i \in I} \bigvee_{j \in J_i} f_j$$

with $I, J_i \subset \mathbb{N}$ and $f_j \in R$.

Garrett Birkhoff demonstrated $\ell$-rings can have surprising pathologies; for example, he constructed an $\ell$-ring whose multiplicative identity was both negative and a square (Birkhoff & Pierce, 1956). The following definition introduces an additional compatibility condition that prevents the most troublesome pathologies.

**Definition 15.** An $f$-*ring* (short for "function ring") is a $\ell$-ring where

$$\text{If } f, g, h \geq 0 \text{ and } g \wedge h = 0, \text{ then } fg \wedge h = 0 \text{ and } gf \wedge h = 0. \tag{18}$$

Condition (18) can be equivalently stated (Madden, 2011)

$$f^+ g^+ \wedge f^- = 0 \quad \text{and} \quad g^+ f^+ \wedge f^- = 0 \quad \text{for all } f, g \in R.$$

**Lemma 16.** *If $R$ is an $\ell$-ring (resp. $f$-ring) and $D$ is any set, then functions $F = \{f : D \to R\}$ form an $\ell$-ring (resp. $f$-ring) using the operations defined to turn every evaluation map $\phi_x(f) := f(x)$, $x \in D$, into an $\ell$-ring homomorphism (stated simply, these are the "component-wise" operations).*

*Proof.* For brevity, we forego the verification of the ring and lattice axioms and focus on the compatibility conditions. The first will be treated in detail for illustration, the other two will be more succinct. Let $f, g, h \in F$.

---

[1]Only compatibility condition (16) is required, so in fact every so-called $\ell$-*group* is a distributive lattice.

- Condition (16): Suppose $f \leq g$, which means $f(x) \leq g(x)$ for all $x \in D$:

$$f(x) = (f \wedge g)(x) = \phi_x(f \wedge g) = \phi_x(f) \wedge \phi_x(g) = f(x) \wedge g(x).$$

Since $R$ is $\ell$-ring, $a + f(x) \leq a + g(x)$ for all $a \in R$ and $x \in D$. In particular, $h(x) + f(x) \leq h(x) + g(x)$, and thus

$$\phi_x(h + f) = h(x) + f(x) = \big(h(x) + f(x)\big) \wedge \big(h(x) + g(x)\big) =$$
$$\phi_x(h + f) \wedge \phi_x(h + g) = \phi_x\big((h + f) \wedge (h + g)\big)$$

Since $\wedge$ in $F$ is defined precisely to turn every $\phi_x$ into a homomorphism, we conclude $h + f = (h + f) \wedge (h + g)$.

- Condition (17): Suppose $f, g \geq 0$, meaning $f(x), g(x) \geq 0$ for all $x \in D$. Then $f(x)g(x) \geq 0$ since $R$ is an $\ell$-ring.

For the last condition, we now suppose $R$ is an $f$-ring.

- Condition (18): Suppose $f, g, h \geq 0$ and $f \wedge h = 0$, meaning $f(x), g(x), h(x) \geq 0$ and $f(x) \wedge h(x) = 0$ for all $x \in D$. Then

$$(fg \wedge h)(x) = f(x)g(x) \wedge h(x) = 0$$

since $R$ is an $f$-ring, and we conclude $fg \wedge h = 0$. Likewise $gf \wedge h = 0$.

$\square$

**Example 17.** Continuous real-valued functions with component-wise $+/*/$min/max operations form an $f$-ring.

**Lemma 18.** *The direct product of $\ell$-rings (resp. $f$-rings) is an $\ell$-ring (resp. $f$-ring).*

*Proof.* The direct product of two $\ell$-rings or $f$-rings is clearly both a ring and a lattice, and the relevant compatibility conditions hold in the product since they hold in each component. $\square$

In particular, vector-valued functions $D \rightarrow \mathbb{R}^n$ form an $f$-ring when each component forms an $f$-ring.

**Lemma 19.** *If $R$ is an $\ell$-ring (resp. $f$-ring) and $R_1 \subseteq R$ is both a subring and sublattice, then $R_1$ is also an $\ell$-ring (resp. $f$-ring).*

*Proof.* We need only verify the compatibility conditions (16), (17) and (18). But since these conditions hold in $R$, they clearly hold in $R_1$ as well. $\square$

The requirement that $R_1 \subseteq R$ form a sublattice in addition to a subring (rather than just a subring) is a small oversight in (Birkhoff & Pierce, 1956). For example, polynomials form a subring of the $f$-ring of continuous real functions, but they are not a lattice ($x$ and $-x$ are both polynomials, but $x \vee -x = |x|$ is not), so polynomials do not themselves form an $f$-ring. Indeed, this fact will be relevant below to us below.

**Definition 20** (Totally ordered $\ell$ ring)**.** An $\ell$-ring $R$ is *totally ordered* if both $f \wedge g$ and $f \vee g$ are in $\{f, g\}$ for all $f, g \in R$.

In other words, the lattice of a totally ordered ring induces a total order relation.

**Lemma 21.** *Every totally ordered ring $R$ is an $f$-ring.*

*Proof.* Given $f, g, h \in R$ such that $f, g, h \geq 0$ and $g \wedge h = 0$, we need to show $fg \wedge h = 0$ and $gf \wedge h = 0$. Since $R$ is totally ordered, $g \wedge h = 0$ implies either $g = 0$ or $h = 0$.

- Case $g = 0$: We have $fg = gf = 0$ and $fg \wedge h = 0 \wedge h = 0$ since $h \geq 0$. Identical logic shows $gf \wedge h = 0$.

- Case $h = 0$: From the $\ell$-ring compatibility condition (17), $fg \geq 0$ and $gf \geq 0$. Thus $fg \wedge h = fg \wedge 0 = 0$, likewise $gf \wedge h = gf \wedge 0 = 0$.

$\square$

We can now state the key theorem of this section, which will allow us to generate $f$-rings of neural networks by bootstrapping from a subring.

**Theorem 22** (Henricksen and Isbell). *If $R$ is an $f$-ring and $R_1 \subset R$ is a subring (not necessarily a sublattice), then the lattice generated by $R_1$ is a subring of $R$.*

This theorem first appeared in Henriksen & Isbell (1962). The proof is surprisingly non-trivial, and while the key insights were provided by Henricksen and Isbell, they left the proof as an exercise to the reader. The first full proof appears to have been recorded in Hager & Johnson (2010).

## A.1 INF-SUP DEFINABLE PIECEWISE POLYNOMIALS AND THE PIERCE–BIRKHOFF CONJECTURE

**Definition 23** (Piecewise polynomial). *Let $k, n, J \in \mathbb{N}$, and $D \subset \mathbb{R}^n$. Then $f : D \to \mathbb{R}$ is piecewise polynomial, or $f \in PWP(D)$, if there exists a collection of semialgebraic sets $\{S_i\}_{i=1}^J$ such that $D \subset \bigcup_{i=1}^J S_i$, and a collection of polynomials $\{p_i\}_{i=1}^J \subset \mathbb{R}[X_1, \dots, X_n]$ such that $f|_{S_i} = p_i$ for all $i = 1, \dots, J$.*

*A vector-valued function $f : D \to \mathbb{R}^k$ is piecewise polynomial, or $f \in PWP(D, \mathbb{R}^k)$, if every component function is piecewise polynomial.*

A piecewise polynomial is continuous if and only if there exists a collection of *closed* semialgebraic sets $\{S_i\}_{i=1}^J$ satisfying Definition 23.

**Lemma 24.** *Let $n \in \mathbb{N}$, and $D \subset \mathbb{R}^n$. Then $PWP(D)$ equipped with component-wise operations is an $f$-ring.*

*Proof.* We need only show $PWP(D)$ is both a subring and sublattice of the $f$-ring of functions $D \to \mathbb{R}$, then the conclusion follows from Lemma 16. But it is elementary to verify the pointwise sum, product, min and max of two piecewise polynomials is also a pointwise polynomial. $\square$

**Corollary 25.** *Let $k, n \in \mathbb{N}$, and $D \subset \mathbb{R}^n$. Then $PWP(D, \mathbb{R}^k)$ is an $f$-ring.*

*Proof.* $PWP(D, \mathbb{R}^k)$ is a direct product of $PWP(D, \mathbb{R})$, so the conclusion follows from Lemma 18. $\square$

We apply Theorem 22 to the subring of $PWP(D, \mathbb{R}^k)$ consisting of the polynomials $D \to \mathbb{R}^k$.

We obtain the following important theorem:

**Theorem 26.** *$ISD(\mathbb{R}^n)$ is an $f$-ring for any $n \in \mathbb{N}$.*

Both $PWP(\mathbb{R}^n)$ and $ISD(\mathbb{R}^n)$ are $f$-rings, and clearly $ISD(\mathbb{R}^n) \subseteq PWP(\mathbb{R}^n)$. Whether the opposite inclusion holds is the famous Pierce–Birkhoff conjecture.

**Conjecture 27** (Pierce–Birkhoff). *$ISD(\mathbb{R}^n) = PWP(\mathbb{R}^n)$.*

If the Pierce–Birkhoff conjecture holds, then every piecewise polynomial $g$ can be written

$$g = \min_{i \in I} \max_{j \in J_i} f_{ij}$$

for some polynomials $f_{ij}$ and finite indexing sets $I, J_i$.

The case $n = 2$ was proved by Jacek Bochnak and Gustave Efroymson (Bochnak et al., 1998), and then again by Louis Mahé (Mahé, 1984). Mahé has also shown that the conjecture holds when $n = 3$ up to arbitrarily small neighborhoods of finitely many points. See Madden (2011) for further discussion of the Pierce–Birkhoff conjecture and related problems.

Later, we will show that the neural network architecture introduced in the next section can exactly compute any continuous piecewise polynomial if and only if the Pierce–Birkhoff conjecture holds.

# B    MATRIX-RECURRENT NEURAL NETWORKS

Semialgebraic Neural Networks (SANNs) are built on top of a related architecture called Matrix-Recurrent Neural Networks (MRNNs), a type of polynomial network with ReLU activations. This chapter introduces MRNNs and exactly characterize their range as inf-sup definable (ISD) piecewise polynomials. Our analysis of the expressivity of MRNNs is inspired by Balestriero & Baraniuk (2021).

## B.1    ARCHITECTURE

A standard feed-forward neural network is a function $f : \mathbb{R}^n \to \mathbb{R}^k$ defined as an alternating composition of affine transformations and nonlinear activation functions $\sigma$:

$$h_0 = x \tag{19}$$
$$h_\ell = \sigma\left(b_\ell + A_\ell h_{\ell-1}\right) \qquad \text{for } \ell = 1, \dots, L-1 \tag{20}$$
$$f(x) = b_L + A_L h_{L-1}. \tag{21}$$

Perhaps the most popular activation functions used in practice are variations of the rectified linear unit (ReLU) $\sigma(x) = \max(0, x)$. This activation results in networks whose ranges are conceptually simple; they are multivariate linear splines. Although such networks can uniformly approximate any continuous function, they are extremely limited in the types of functions that can be represented exactly. For example, the product of two linear splines is in general not a linear spline itself, so computing such a function by a neural network will require a number of weights proportional to the desired accuracy of the approximation.

In this work, we study a type of *Operator Recurrent Neural Network* (ORNN) (de Hoop et al., 2022) where the input to the network is a matrix $X \in \mathbb{R}^{n \times m}$ and the output is a vector $f(X) \in \mathbb{R}^{k_{\text{out}}}$. To emphasize that our results hold specifically for matrix inputs rather than general operators, we call this subset of ORNN's the *Matrix Recurrent Neural Networks* (MRNNs). The architecture of an $L$-layer MRNN is defined

$$h_0 = 0 \tag{22}$$
$$h_\ell = b_{\ell,0} + A_{\ell,0} h_{\ell-1} + B_{\ell,0}(I \otimes X) h_{\ell-1} +$$
$$\qquad \sigma\left(b_{\ell,1} + A_{\ell,1} h_{\ell-1} + B_{\ell,1}(I \otimes X) h_{\ell-1}\right) \qquad \text{for } \ell = 1, \dots, L \tag{23}$$
$$f(X) = h_L \tag{24}$$

where $(I \otimes X)$ denotes a Kronecker product with an identity matrix $I$ whose size may vary in each layer. Notice in particular that the input matrix $X$ is inserted *multiplicatively* into each layer. Our definition of an MRNN matches the original definition of a width-expanded ORNN with matrix input presented in de Hoop et al. (2022). As noted in that paper, the restriction on $h_0$ does not affect the range of $f$ since $h_1$ will invariably equal $b_{1,0}$, which is learned.

We formally define the space of functions representable by an MRNN.

**Definition 28** (MRNN). Let $k, L \in \mathbb{N}$, and $D$ be a subset of $\mathbb{R}^{m \times n}$. Then $f : D \to \mathbb{R}^k$ is in $MRNN_L(D, \mathbb{R}^k)$ if $f$ can be computed via equations (22-24).

When the number of layers in a network is not specified, we define

$$MRNN(D, \mathbb{R}^k) = \bigcup_{L=0}^{\infty} MRNN_L(D, \mathbb{R}^k).$$

In contrast to standard ReLU networks, we will show that MRNNs form a ring of piecewise polynomial functions. The kernels of these functions—that is, the set of points $X$ such that $f(X)$ vanishes—form semialgebraic sets in $\mathbb{R}^{m \times n}$.

## B.2    RANGE

Numerous authors have observed that linear ReLU networks compute multivariate linear splines, where each linear region is a polyhedron defined by the combination of active neurons—that is,

where $b_\ell + A_\ell h_{\ell-1} \geq 0$ (see, for example, Montúfar et al. (2014) and Wang et al. (2019)). Likewise, de Hoop et al. (2022) showed that MRNNs (or rather ORNNS in general) compute piecewise polynomials, where the semialgebraic decomposition of the domain is similarly defined by the combination of active neurons

$$b_{\ell,1} + A_{\ell,1}h_{\ell-1} + B_{\ell,1}(I \otimes X)h_{\ell-1} \geq 0, \qquad \ell = 1, \ldots, L$$

at each point. Thus $MRNN(\mathbb{R}^{n \times m}, \mathbb{R}^k)$ is isomorphic to a subset of $PWP(\mathbb{R}^{nm}, \mathbb{R}^k)$. The converse question, whether every piecewise polynomial is computable by an MRNN, was not addressed. In this subsection, we refine their observation to show that MRNNs compute precisely the ISD functions.

Let $\text{vec} : \mathbb{R}^{n \times m} \to \mathbb{R}^{nm}$ be the column-major matrix vectorization operator. It is clearly bijective, and its inverse is the matrixization operator denoted $\text{vec}^{-1}$. We first show every ISD function can be computed by and MRNN.

**Lemma 29.** *Let $m, n, \in \mathbb{N}$. For every $f \in ISD(\mathbb{R}^{mn})$, there exists $\mathcal{N} \in MRNN(\mathbb{R}^{m \times n})$ such that $f = \mathcal{N} \circ \text{vec}^{-1}$.*

*Proof.* By definition of $ISD(\mathbb{R}^{mn})$, $f$ is a lattice polynomial $q$ of some $\{p_1, \ldots, p_J\} \subset \mathbb{R}[X_1, \ldots, X_{mn}]$. An easy construction shows there exists networks $\mathcal{N}_1, \ldots, \mathcal{N}_J$ computing the polynomials $\mathcal{N}_j = p_j \circ \text{vec}^{-1}$. Another easy construction shows that $MRNN(\mathbb{R}^{m \times n})$ is closed under the lattice operations, so the lattice polynomial $q$ can be computed by a network. In particular, $\mathcal{N} = q(\mathcal{N}_1, \ldots, \mathcal{N}_J)$ is the desired network. $\qquad\square$

We now generalize the previous lemma to vector-valued functions.

**Lemma 30.** *Let $m, n, k \in \mathbb{N}$. For every $f \in ISD(\mathbb{R}^{mn}, \mathbb{R}^k)$, there exists $\mathcal{N} \in MRNN(\mathbb{R}^{m \times n}, \mathbb{R}^k)$ such that $f = \mathcal{N} \circ \text{vec}^{-1}$.*

*Proof.* $ISD(\mathbb{R}^{mn}, \mathbb{R}^k)$ is the $k$-times direct product of $ISD(\mathbb{R}^{mn})$, so $f = f_1 \times \cdots \times f_k$ for some $f_1, \ldots, f_k \in ISD(\mathbb{R}^{mn})$. By the previous lemma, there exist $\mathcal{N}_1, \ldots, \mathcal{N}_k \in MRNN(\mathbb{R}^{m \times n})$ such that $f_j = \mathcal{N}_j \circ \text{vec}^{-1}$ for $j = 1, \ldots, k$. Since $MRNN(\mathbb{R}^{m \times n}, \mathbb{R}^k)$ is the $k$-times direct product of $MRNN(\mathbb{R}^{m \times n})$, $\mathcal{N} = \mathcal{N}_1 \times \cdots \times \mathcal{N}_k$ is the desired network. $\qquad\square$

The next two lemmas prove the dual inclusion—every MRNN is indeed ISD.

**Lemma 31.** *Let $m, n \in \mathbb{N}$. For every $\mathcal{N} \in MRNN(\mathbb{R}^{m \times n})$, there exists $f \in ISD(\mathbb{R}^{mn})$ such that $\mathcal{N} = f \circ \text{vec}$.*

*Proof.* Fix a network computing $\mathcal{N}$, and use induction on the number of layers $L$. The base case $L = 0$ is trivial since a 0 layer network is a constant. Assume the condition holds for $L - 1$ layers, and consider the final layer in the network computing $\mathcal{N}(X)$:

$$
\begin{aligned}
h_L = {}& b_{L,0} + A_{L,0}h_{L-1} + B_{L,0}(I \otimes X)h_{L-1} + \\
& \sigma(b_{L,1} + A_{L,1}h_{L-1} + B_{L,1}(I \otimes X)h_{L-1}).
\end{aligned}
\tag{25}
$$

Notice $h_{L-1}$ is a vector-valued function of $X$ and can be decomposed into its component functions

$$h_{L-1} = g_1 \times \cdots \times g_N$$

for some $N \in \mathbb{N}$ corresponding to the width of this layer. For every $j = 1, \ldots, N$, $g_j(X)$ is computed by an $L - 1$ network, so the inductive hypothesis states there exists $f_j \in ISD(\mathbb{R}^{mn})$ such that $g_j = f_j \circ \text{vec}$. We now rewrite each term of equation (25) using these $f_j$. Let $i \in \{0, 1\}$.

- The constant terms $b_{L,i} \in \mathbb{R}$ are clearly isomorphic to constant functions in $ISD(\mathbb{R}^{mn})$.

- The linear terms $A_{L,i}h_{L-1}$ are linear combinations of $f_1, \ldots, f_N \in ISD(\mathbb{R}^{mn})$. Since $ISD(\mathbb{R}^{mn})$ is an $f$-ring (Theorem 26), there exists $f_{A,i} \in ISD(\mathbb{R}^{mn})$ such that $f_{A,i} = (A_{L,i}h_{L-1}) \circ \text{vec}$.

- For the quadratic terms, note $(I \otimes X) \in \mathbb{R}^{rm \times N}$. The $k$'th component of $(I \otimes X)h_{L-1}$ equals

$$\sum_{\alpha=1}^{m} X_{(k\%n,\alpha)} g_{(\lfloor k/n \rfloor + \alpha)} = \left( \sum_{\alpha=1}^{m} X_{(k\%n,\alpha)} f_{(\lfloor k/n \rfloor + \alpha)} \right) \circ \mathrm{vec}(X) \qquad (26)$$

where $\%$ denotes modular division, $\lfloor \cdot \rfloor$ is the floor operation, and $X_{(a,b)}$ is entry $a, b$ of matrix $X$. In particular, each term in the sum on the right-hand-side is the product of a monomial and an $ISD$ function, so

$$f_{B,i,k} := \sum_{\alpha=1}^{m} X_{(k\%n,\alpha)} f_{(\lfloor k/n \rfloor + \alpha)}$$

is in $ISD(\mathbb{R}^{mn})$ for $k = 1, \ldots, rm$. The full quadratic terms $B_{L,i}(I \otimes X)h_{L-1}$ are thus isomorphic to linear combinations of ISD functions $f_{B,i,1}, \ldots, f_{B,i,rm}$, so there exists $f_{B,i} \in ISD(\mathbb{R}^{mn})$ such that $f_{B,i} = (B_{L,i}(I \otimes X)h_{L-1}) \circ \mathrm{vec}$.

We can thus rewrite (25) in the equivalent form

$$h_L = \left[ b_{L,0} + f_{A,0} + f_{B,0} + \sigma(b_{L,1} + f_{A,1} + f_{B,1}) \right] \circ \mathrm{vec}. \qquad (27)$$

The term in square brackets is constructed from elements of the $f$-ring $ISD(\mathbb{R}^{mn})$ using only addition and the lattice operation $\sigma = \max\{0, \cdot\}$, so it is itself in $ISD(\mathbb{R}^{mn})$ and the lemma is proved. $\qquad \square$

**Lemma 32.** *Let* $m, n, k \in \mathbb{N}$. *For every* $\mathcal{N} \in MRNN(\mathbb{R}^{m \times n}, \mathbb{R}^k)$, *there exists* $f \in ISD(\mathbb{R}^{mn}, \mathbb{R}^k)$ *such that* $\mathcal{N} = f \circ \mathrm{vec}$.

*Proof.* We pass through the direct products in an identical manner to the proof of Lemma 30. $\qquad \square$

Combining Lemmas Lemma 30 and Lemma 32 lets us conclude $MRNN(\mathbb{R}^{n \times m}, \mathbb{R}^k)$ is isomorphic to $ISD(\mathbb{R}^{nm}, \mathbb{R}^k)$; that is, there exists a bijective ring-homomorphism between them. Informally, the two rings are identical up to a relabeling of the elements.

**Theorem 33.** *For all* $n, m, k \in \mathbb{N}$, *the rings* $MRNN(\mathbb{R}^{n \times m}, \mathbb{R}^k)$ *and* $ISD(\mathbb{R}^{nm}, \mathbb{R}^k)$ *are homeomorphic, that is,*

$$MRNN(\mathbb{R}^{n \times m}, \mathbb{R}^k) \cong ISD(\mathbb{R}^{nm}, \mathbb{R}^k).$$

Theorems 33 and 22 allow us to conclude that $MRNN(\mathbb{R}^{n \times m}, \mathbb{R}^k)$ forms a unital, associative algebra over $\mathbb{R}$; that is, every scalar multiple, sum, and Hadamard product of a MRNNs is computable by an MRNN. Associativity and existence of a multiplicative identity, as well as the ability to compute scalar multiples and sums are obvious and hold for linear ReLU networks as well. What distinguishes MRNNs from linear ReLU networks is the ability to compute Hadamard products. We require this property to build SANNs capable of computing any semialgebraic function.

While the proofs above are not explicitly constructive, they can be made constructive using the identities in Henriksen & Isbell (1962) or Hager & Johnson (2010). Unfortunately, representing the Hadamard product of two MRNNs as an MRNN requires exponentially many parameters, which limits the practical usefulness of a direct application of these constructions in code.

In their original paper introducing the architecture, de Hoop et al. (2022) note that the range of an MRNN is a multivariate piecewise polynomial; we have refined this result to show that MRNNs are ISD. We now ask the natural question, is every piecewise polynomial representable by an MRNN? As noted previously, this question is the famous Pierce–Birkhoff conjecture, a long-standing open problem in real algebraic geometry. MRNNs can represent every piecewise polynomials if and only if the Pierce–Birkhoff conjecture is true.

## C  SEMIALGEBRAIC FUNCTIONS AS KERNELS OF ISD FUNCTIONS

This appendix contains proofs of Proposition 7 and Corollary 8 from the main text.

**Proposition 7.** $S \subset \mathbb{R}^m$ *is a closed semialgebraic set if and only if there exists* $f \in ISD^1(\mathbb{R}^m, \mathbb{R}_{\geq 0})$ *such that* $\ker(f) = S$.

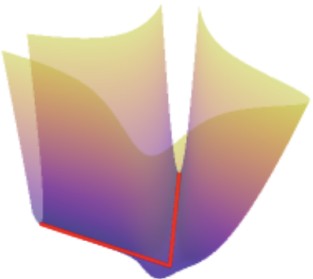

Figure 4: Encoding the graph of a semialgebraic function (red line) $F(x) = |x|$ as the kernel of a continuously differentiable piecewise polynomial (purple and yellow surface)

$$G(x, y) = \big((x + y)(x - y)\big)^2 + \max(0, -y)^2.$$

*Proof.* Clearly the kernel of a piecewise polynomial $f$ is a closed semialgebraic set, so we focus on the converse.

First, suppose $S$ is a closed basic semialgebraic set described by the system

$$p_1(x), \ldots, p_{J_1}(x) = 0 \text{ and } q_1(x), \ldots, q_{J_2}(x) \geq 0.$$

The desired ISD piecewise polynomial is

$$f(x) = \sum_{i=1}^{J_1} p_i^2(x) + \sum_{i=1}^{J_2} \Big( \max\{0, -q_i(x)\} \Big)^2.$$

Each term $p_i^2(x)$ (resp. $\max\{0, -q_i(x)\}^2$) is zero precisely when the corresponding condition $p_i(x) = 0$ (resp. $q_i(x) \geq 0$) is satisfied. Also, the function $q \mapsto \max\{0, q\}^2$ is in $C^1$ and has the continuous derivative $q \mapsto 2\max\{0, q\}$. Moreover, each term is non-negative, so the sum is non-negative and zero if and only if every term is zero, meaning every condition is satisfied simultaneously. Finally, $f$ is ISD since it is constructed using the ISD lattice-ring operations on polynomials (see Appendix A), and it is $C^1$ since each term in the sum is $C^1$.

Now let $S$ be an arbitrary closed semialgebraic set; i.e., the union of basic semialgebraic sets. We may take the basic sets to be closed, since

$$\mathrm{cl}\left(\bigcup_{i=1}^n s_i\right) = \bigcup_{i=1}^n \mathrm{cl}(s_i)$$

for finitely many sets $s_1, \ldots, s_n$. Since, $\ker(fg) = \ker(f) \cup \ker(g)$ for any real-valued functions $f$ and $g$, the desired ISD function can be constructed as the product of piecewise polynomials corresponding to each closed basic semialgebraic set. □

**Corollary 8.** $F : \mathbb{R}^m \to \mathbb{R}^n$ *is a semialgebraic function with closed graph if and only if there exists a* $G \in ISD^1(\mathbb{R}^m \times \mathbb{R}^n, \mathbb{R}^n_{\geq 0})$ *such that* $\ker(G) = gr(F)$, *where* $gr(F) := \{(x, F(x))\} \subset \mathbb{R}^m \times \mathbb{R}^n$ *is the graph of* $F$.

*Proof.* If $F$ is semialgebraic and has a closed graph, then Proposition 7 guarantees there exists a $C^1$ ISD piecewise polynomial $g : \mathbb{R}^m \times \mathbb{R}^n \to \mathbb{R}_{\geq 0}$ such that $gr(F) = \ker(g)$. An $\mathbb{R}^n$-valued function $G$ can be obtained by defining each component function to be $g$.

Now suppose for a given $F$, there exists an ISD $G$ such that $\ker(G) = gr(F)$. Then $gr(F)$ is closed since kernels of continuous functions are closed, and $F$ is a semialgebraic function since $\ker(G)$ is clearly a semialgebraic set. □

*Remark.* The hypothesis in Corollary 8 that $F$ has a closed graph covers all continuous semialgebraic functions, as well as some unbounded discontinuous functions like

$$y = \begin{cases} 0, & x \leq 0 \\ 1/x, & x > 0. \end{cases} \tag{28}$$

For an example of a semialgebraic function whose graph is not closed, consider the characteristic function of $(0, +\infty)$:

$$y = \begin{cases} 0, & x \leq 0 \\ 1, & x > 0. \end{cases} \tag{29}$$

The graph does not contain the limit point $(0, 1)$.

## D  HOMOTOPY CONTINUATION METHODS

This appendix contains additional background material on homotopy continuation methods for root finding. Subsection D.1 states semialgebraic variants of two important theorems from geometry that are the foundation for our proofs. Subsection D.2 gives sufficient conditions for the existence of a homotopy $H$ whose kernel can be traced from time $0$ to time $1$ in order evaluate the roots of a given function. Finally, D.3 builds the system of ODEs that is solved in order to execute the homotopy continuation method, and we prove this ODE system can be represented by a SANN.

### D.1  SEMIALGEBRAIC SARD'S AND TRANSVERSALITY THEOREMS

We present semialgebraic variations on two classic theorems from differential topology.

**Theorem 34** (Semialgebraic Sard's theorem). *The set of critical values of a continuously differentiable semialgebraic function has measure zero.*

In the "textbook" Sard's theorem, the smoothness requirements on the function depend on the dimension of the range (Abraham & Robbin, 1967). In contrast, for semialgebraic functions continuous differentiability is sufficient for the theorem to hold. A proof can be found in Kurdyka et al. (2000).

Next, we use the previous theorem to adapt the Thom transversality theorem to $C^1$ semialgebraic functions.

**Theorem 35** (Semialgebraic transversality theorem). *If $F : X \times A \to Y$ is continuously differentiable semialgebraic function and $0$ is a regular value of $F$, then $0$ is a regular value of $F(\cdot, a)$ for almost every $a \in A$.*

A proof can be found in (Abraham & Robbin, 1967), substituting the semialgebraic Sard's theorem for Smale's theorem as required.

### D.2  HOMOTOPY EXISTENCE

We now state a standard result in the theory of homotopy continuation for solving nonlinear equations. Theorem 37 is adapted from Krantz & Parks (2003) with small modifications using the theorems from Appendix D.1 to specialize to $ISD^1$ and avoid any $C^\infty$ assumptions. It provides sufficient conditions under which we are guaranteed to be able to evaluate a homotopy from $t = 0$ until $t = 1$.

**Definition 36** (Regular value). Let $U \subset \mathbb{R}^m$ be open. Given $f : U \to \mathbb{R}^n$, $y \in \mathbb{R}^n$ is a *regular value* of $f$ if the Jacobian matrix $Df(x)$ has rank $n$ for all $x \in f^{-1}(y)$.

In particular, $0$ is a regular value of $f : \mathbb{R}^n \to \mathbb{R}^n$ if and only if the Jacobian matrix is nonsingular for every $x \in \ker(f)$.

**Theorem 37.** *Let $U_H$ be a bounded open subset of $\mathbb{R}^n$, and $H \in ISD^1(\mathbb{R}^n \times \mathbb{R}, \mathbb{R}^n)$. Further suppose*

    *A.1  $H(\cdot, 0) = 0$ has a unique solution $y_0$ in $U_H$. That is, $\ker H(\cdot, 0) \cap U_H = \{y_0\}$.*

    *A.2  The connected component of $\ker H$ containing $(y_0, 0)$ does not intersect $\partial U_H \times [0, 1]$.*

**A.3** $0$ *is a regular value of $H$.*

**A.4** $\partial_1 H(y_0, 0)$ *is nonsingular.*

*Then there exists $t \in ISD([0,1], \mathbb{R})$, $y \in ISD([0,1], \mathbb{R}^n)$ such that $t(0) = 0$, $t(1) = 1$, and $(y(s), t(s)) \in \ker(H)$ for all $s \in [0,1]$.*

### D.3 Curve Tracing

Let $z(s) := \big(y(s), t(s)\big)$, and $H \in ISD^1(\mathbb{R}^n \times \mathbb{R}, \mathbb{R}^n)$ satisfy the hypotheses of Theorem 37. Then $(y, 1) \in \ker H$ can be computed by solving the ODE initial-value problem defined by the "arc-length parameterization" (Chen & Li, 2015):

$$DH(z) \cdot \dot{z} = 0 \tag{30}$$

$$\text{sgn} \det \begin{bmatrix} DH(z) \\ \dot{z} \end{bmatrix} = \sigma_0 \tag{31}$$

$$\|\dot{z}\| = \beta \tag{32}$$

$$z(0) = (y_0, 0). \tag{33}$$

where $\sigma_0$ is the sign of the determinant of that matrix at $z(0)$ and $\beta > 0$ is a constant chosen so that $t(1) = 1$.

With proper choice of $\mathcal{N}$ and $c_{\max}$, the curve-tracing ODE system (30)–(33) can be written in the form of the ODE system (6)–(7) that define SANNs. At time $s = 0$, write

$$DH(y_0, 0) = \left[\; \widehat{M} \;\middle|\; \widehat{b} \;\right]. \tag{34}$$

That is, $\widehat{M}$ is the $n \times n$ submatrix of $DH(y_0, 0)$ that does not contain the rightmost columns $\widehat{b}$. By assumption **A.4**, $\widehat{M}$ is invertible. Thus a vector $\dot{z}$ satisfying (30)–(33) can be obtained by solving

$$M\dot{z} := \begin{bmatrix} \widehat{M} & 0 \\ 0 & \alpha \end{bmatrix} \dot{z} = \begin{bmatrix} \widehat{b} \\ 1 \end{bmatrix} =: b \tag{35}$$

where $\alpha$ is a constant chosen to satisfy (31) and (32). The $M$ and $b$ defined above appear in the ODE (6)–(7). Thus we have shown for $H$ satisfying the hypotheses of Theorem 37, $y_0 = 0$, and large enough $c_{\max}$, there is an $\mathcal{N} \in ISDnet(m, n, k)$ such that the integral curve traced in solving the ODE (6)–(7) is "correct" at time $s = 0$; i.e. it is tangent to the integral curve defined by $\dot{z}$ solving the IVP (30)–(33).

For time $s > 0$, we are no longer guaranteed that the first $n$ columns of $DH$ form an invertible submatrix. However, by assumption **A.3**, $DH$ always has rank $n$, so every $s$ belongs to some interval where we can choose a column of $DH$ to be $\widehat{b}$, and the remaining columns to be $\widehat{M}$. The computations in each of these intervals can be glued together in a way consistent with the ODE (6)–(7) used to compute the entire integral curve. The details can be found in the proof of Theorem 12 in Appendix E.

## E   SANN Expressivity Proofs

In this appendix, we prove via construction that SANNs are able to represent both continuous and discontinuous bounded semialgebraic functions $F$.

### E.1   Notation

The theorems in this section require us to show the existence of ISD $\mathcal{N}$ generating certain initial value problems for (6)–(7). In Algorithm 1, $\dot{z}$ implicitly depends on the inputs $\mathcal{N}$, $c_{\max}$, and $x$, and `ODESolve` always runs from time $s = 0$ to $s = 1$. For this section, we make the dependence on all variables explicit by using the signature

$$\texttt{ODESolve}(\mathcal{N}, c_{\max}, x, (y_0, t_0, s_0), s_{\text{final}})$$

where $\mathcal{N} \in ISDnet(m, n, 1)$ is the trainable ISD network, $c_{\max} \in \mathbb{R}_{\geq 0}$ bounds the output of the SANN, $x \in \mathbb{R}^m$ is the input to the SANN, $(y_0, t_0, s_0) \in \mathbb{R}^n \times \mathbb{R} \times \bar{\mathbb{R}}$ is the initial condition, and $s_{\text{final}} \in \mathbb{R}$ is the final time. We assume that `ODESolve` exactly solves the ODE system defined by $\dot{z}$ in Algorithm 1 from time $s = s_0$ to time $s = s_{\text{final}}$. To avoid repetition, $\mathcal{N}$, $c_{\max}$, $x$, $y_0$, $t_0$, $s_0$, and $s_{\text{final}}$ will always be in their respective spaces above. Furthermore, for $\mathcal{N}_\alpha$ with any subscript $\alpha$, we use $M_\alpha$ and $b_\alpha$ to be the associated projections as in line 3 of Algorithm 1.

We use $\Pi_{n+1}$ to denote the projection operator selecting $t_{\text{out}}$ from $(y_{\text{out}}, t_{\text{out}})$.

### E.2 GLUEING

We require two different ways to combine ISD networks $\mathcal{N}_1$ and $\mathcal{N}_2$. For "$s$-glueing," we build $\mathcal{N}_3$ that first imitates $\mathcal{N}_1$ on an interval $s \in [s_0, s_{1/2}]$, then behaves as $\mathcal{N}_2$ when $s \in [s_{1/2}, s_{\text{final}}]$. We require an additional lemma that modifies $\mathcal{N}_1$ and $\mathcal{N}_2$ to guarantee $\dot{y}, \dot{t}$ can be forced to be 0 in an open set around $s_{1/2}$ without affecting the output of `ODESolve`. We accomplish this with an ISD change of variables.

For "$t$-glueing," we build $\mathcal{N}_3$ to interpolate between $\mathcal{N}_1$ and $\mathcal{N}_2$ in disjoint closed $t$ intervals $[t_0, t_1]$ and $[t_2, t_3]$. We use $t$-glueing when we can guarantee the integral curve traced by `ODESolve` will never enter the region $(t_1, t_2)$, so the value of $\mathcal{N}_3$ may be arbitrary here. We are free to use this interval to interpolate between $\mathcal{N}_1$ and $\mathcal{N}_2$.

Our first lemma allows a particular change of variables for $s$-glueing.

**Lemma 38** (Change-of-variables). *Given $\mathcal{N}_1$, $c_1$, $y_0$, $t_0$, and $s_0 < s_1 < s_2 < s_{\text{final}}$, there exists $\mathcal{N}_2, c_2$ such that for all $x$,*

$$\text{ODESolve}(\mathcal{N}_1, c_1, x, (y_0, t_0, s_0), s_{\text{final}}) = \text{ODESolve}(\mathcal{N}_2, c_2, x, (y_0, t_0, s_0), s_{\text{final}}) \qquad (36)$$

*and*

$$b_2(x, y, t, s) = 0 \quad \text{when } s \notin [s_1, s_2]. \qquad (37)$$

*Proof.* Let $v : \mathbb{R} \to \mathbb{R}$ be defined

$$v(s) = \begin{cases} s_0 & s < s_1 \\ u(s) & s \in [s_1, s_2] \\ s_{\text{final}} & s > s_2 \end{cases} \qquad (38)$$

where $u$ is the Hermite interpolant (Süli & Mayers, 2003) such that

$$u(s_1) = s_0 \qquad\qquad \dot{u}(s_1) = 0 \qquad (39)$$
$$u(s_2) = s_{\text{final}} \qquad\qquad \dot{u}(s_2) = 0. \qquad (40)$$

$v$ is continuously differentiable and $\dot{v} \in ISD(\mathbb{R})$, $\dot{v}(s) = 0$ for all $s \notin [s_1, s_2]$. Since $s \mapsto \dot{v}(s)b(x, z(s))$ is the product of ISD functions, it is itself ISD. Likewise the composition of an ISD function with a polynomial is ISD. Thus we can define

$$M_2(x, y, t, s) = M_1(x, y(v(s)), t(v(s)), v(s)) \qquad (41)$$
$$b_2(x, y, t, s) = \dot{v}(s)b_1(x, y(v(s)), t(v(s)), v(s)). \qquad (42)$$

Simple calculus verifies conclusion (36), and conclusion (37) holds since $\dot{v}(s) = 0$ outside $[s_1, s_2]$. $\qquad \square$

**Lemma 39** (*s*-glueing). *Given $\mathcal{N}_1$, $\mathcal{N}_2$, $c_1$, $c_2$ $y_0$, $t_0$, $s_0 < s_1 < s_{\text{final}}$, let*

$$(y_{out}(x), t_{out}(x)) = \text{ODESolve}(\mathcal{N}_1, c_1, x, (y_0, t_0, s_0), s_1). \qquad (43)$$

*There exists $\mathcal{N}_3$, $c_3$ such that for all $x$,*

$$\text{ODESolve}(\mathcal{N}_3, c_3, x, (y_0, t_0, s_0), s_{\text{final}}) = \text{ODESolve}(\mathcal{N}_2, c_2, x, (y_{out}(x), t_{out}(x), s_1), s_{\text{final}}). \qquad (44)$$

*Proof.* Choose $\epsilon > 0$ such that $s_0 < s_1 - \epsilon < s_1 + \epsilon < s_{\text{final}}$, and apply Lemma 38 to $\mathcal{N}_1$ and $\mathcal{N}_2$ so that both $b_1$ and $b_2$ are identically 0 in the region $(s_1 - \epsilon, s_1 + \epsilon)$. No matter the value of $M_1$ and $M_2$, both $\dot{y}$ and $\dot{t}$ are identically 0 here. Let $\lambda(s) = (s - (s_1 - \epsilon))/(2\epsilon)$. Define

$$
M_3(x, y, t, s) = \begin{cases} M_1(x, y, t, s) & s \le s_1 - \epsilon \\ (1 - \lambda(s))M_1(x, y, t, s) + \lambda(s)M_2(x, y, t, s) & s \in (s_1 - \epsilon, s_1 + \epsilon) \\ M_2(x, y, t, s) & s \ge s_1 + \epsilon \end{cases} \tag{45}
$$

and

$$
b_3(x, y, t, s) = \begin{cases} b_1(x, y, t, s) & s \le s_1 \\ b_2(x, y, t, s) & s > s_1. \end{cases} \tag{46}
$$

$M_3$ is clearly continuous, and $b_3$ is continuous since $b_1$ and $b_2$ are both 0 when $s = s_1$. The constructed $\mathcal{N}_3$ is ISD and satisfies conclusion (44). □

*Remark.* Strictly speaking, we have not written $M_3$ and $b_3$ in ISD form in the proof above, but this is not difficult to remedy. For example, we could use

$$
\lambda(s) := \min(\max((s - (s_1 - \epsilon))/(2\epsilon), 0), 1), \tag{47}
$$

then define

$$
M_3 = (1 - \lambda(s))M_1 + \lambda(s)M_2 \tag{48}
$$
$$
b_3 = (1 - \lambda(s))b_1 + \lambda(s)b_2 \tag{49}
$$

over the entire domain.

**Lemma 40** ($t$-glueing). *Given $\mathcal{N}_1$, $\mathcal{N}_2$, $c_1$, $c_2$, $y_1$, $y_2$, $t_1$, $t_2$, $s_0$, $s_{\text{final}}$, let*

$$
\mathcal{T}_j := \{\Pi_{n+1}\texttt{ODESolve}(\mathcal{N}_j, c_j, x, (y_j, t_j, s_0), s) \mid s \in [s_0, s_{\text{final}}], x \in \mathbb{R}^m\}. \tag{50}
$$

*Suppose $\mathcal{T}_1 \cap \mathcal{T}_2 = \emptyset$. Then there exists $\mathcal{N}_3$, $c_3$ such that*

$$
\texttt{ODESolve}(\mathcal{N}_3, c_3, x, (y_j, t_j, s_0), s_{\text{final}}) = \texttt{ODESolve}(\mathcal{N}_j, c_j, x, (y_j, t_j, s_0), s_{\text{final}}) \tag{51}
$$

*for $j = 1, 2$ and $x \in \mathbb{R}^m$.*

*Proof.* Sets $\mathcal{T}_j$ contain the integral curves for the $t$ output of $\texttt{ODESolve}$, so they are connected and closed. Since they are disjoint, we can assume WLOG $\max \mathcal{T}_1 < \min \mathcal{T}_2$. Then there exists $t_{\text{mid}}, \epsilon \in \mathbb{R}$ such that

$$
\max \mathcal{T}_1 < t_{\text{mid}} - \epsilon < t_{\text{mid}} + \epsilon < \min \mathcal{T}_2. \tag{52}
$$

The integral curves traced by $\texttt{ODESolve}$ do not enter the $t$-region $(t_{\text{mid}} - \epsilon, t_{\text{mid}} + \epsilon)$, so we are free to define $\mathcal{N}_3$ to interpolate between $\mathcal{N}_1$ and $\mathcal{N}_2$ there. Let $\lambda(t) = (t - (t_{\text{mid}} - \epsilon))/(2\epsilon)$. Define

$$
M_3 = \begin{cases} M_1 & t \le t_{\text{mid}} - \epsilon \\ \lambda(t)M_1 + (1 - \lambda(t))M_2 & t \in (t_{\text{mid}} - \epsilon, t_{\text{mid}} + \epsilon) \\ M_2 & t \ge t_{\text{mid}} - \epsilon \end{cases} \tag{53}
$$

$$
b_3 = \begin{cases} b_1 & t \le t_{\text{mid}} - \epsilon \\ \lambda(t)b_1 + (1 - \lambda(t))b_2 & t \in (t_{\text{mid}} - \epsilon, t_{\text{mid}} + \epsilon) \\ b_2 & t \ge t_{\text{mid}} - \epsilon \end{cases} \tag{54}
$$

$$
c_3 = \max\{c_1, c_2\}. \tag{55}
$$

□

### E.3 PROOF OF THEOREM 11

**Theorem 11.** *Let $F : \mathbb{R}^m \to U_F$ be a given continuous semialgebraic function. For every $x \in \mathbb{R}^m$ there exists $H_{x,a}$ with the form (10) such that for almost every $a \in U_F$, the following hold:*

1. *There exists $y \in ISD([0, 1], \mathbb{R}^n), t \in ISD([0, 1], \mathbb{R})$, such that $t(0) = 0$, $t(1) = 1$, and $(y(s), t(s)) \in \ker(H_{x,a})$ for all $s \in [0, 1]$.*

2. *The kernel of $H_{x,a}(\cdot, 1)$ is the singleton $\{F(x)\}$.*

*Proof.* To specify $H_{x,a}$, we need to choose $g$ and $G$. Let $g$ be constructed according to Lemma 10. Let $G$ be constructed such that $\ker G = \text{gr}(F)$ using Corollary 8.

For conclusion 1, we will use Theorem 37. We first specify a bounded, open $U_H \subset \mathbb{R}^n$ containing $U_F$ with an additional property that is relevant for **A.2**. Let $\mathcal{C}$ denote the connected component of $\ker(H_{x,a})$ containing $(y_0, 0)$ (i.e. the curve to be traced), and assume for the moment **A.3** (which does not depend on choice of $U_H$) has been verified. Then for almost every $a$, there exists $t_0 \in (0, 1)$ such that $\mathcal{C} \cap (\mathbb{R}^n \times [0, t_0]) = \mathcal{C} \cap (U_F \times [0, t_0])$; in other words, the curve remains in $U_F$ until at least time $t_0$. Now, using Lemma 10, we conclude there exists bounded open $U_H$ containing $U_F$ such that $H_{x,a}(t, y) > 0$ for all $(y, t) \in U_H^c \times [t_0, 1]$.

We now consider each hypothesis of Theorem 37 in turn:

**A.1** The unique solution to $H_{x,a}(y, 0) = y - a = 0$ is $y_0 = a$.

**A.2** From the construction of $U_H$ above, the curve $\mathcal{C}$ does not intersect $\partial U_H$ during the interval $t \in [0, t_0]$, and the entire kernel of $H_{x,a}$ is disjoint from $\partial U_H$ for $t > t_0$.

**A.3** We will show that 0 is a regular value of $H_{x,a}$ for almost every $a$ using the Semialgebraic Transversality Theorem (see Theorem 35 in appendix D.1). Consider $\tilde{H}_x : \mathbb{R}^n \times \mathbb{R} \times \mathbb{R}^n \to \mathbb{R}^n$ obtained from $H_{x,a}$ by treating $a$ as a variable: $\tilde{H}_x(y, t, a) := H_{x,a}(y, t)$. Then

$$\partial_3 \tilde{H}_x = -(1-t)I_n,$$

which is invertible for $t \neq 1$, so 0 is a regular value of $\tilde{H}$. Apply the Theorem 35 to $\tilde{H}_x$ to conclude 0 is a regular value of $H_{x,a}$ for almost every $a$.

**A.4** $(y_0, 0)$ is in the region $U_F \times U_F$, so $\partial_1 H_{x,a}(y_0, 0) = I_n$, which is nonsingular.

For conclusion 2, we have shown in **A.2** that $y(1) \in U_F$ where $g$ vanishes, so $H_{x,a}(y(1), 1) = G(x, y(1))$. The conclusion follows since $\ker G = \text{gr}(F)$. $\qquad\square$

### E.4 Proof of Theorem 12

**Theorem 12.** *Let $F : \mathbb{R}^m \to U_F$ be a given continuous semialgebraic function. Then there exists $\mathcal{N} \in ISDnet(m, n, 1)$ and $c_{\max} > 0$ such that $SANN^{lim}(\mathcal{N}, \cdot, c_{\max}) = F$.*

*Proof.* In Section D.3, we showed how to construct $\mathcal{N}_0 \in ISDnet(m, n, 1)$ to compute the required integral curve using Algorithm 1 at time $s = 0$. $\mathcal{N}_0$ can be chosen in this manner to compute the integral curve until some time $s_1 > 0$, when $\widehat{M}(s_1)$, the first $n$ columns of $DH(y(s_1), t(s_1))$, no longer form an invertible submatrix. However, by assumption **A.3**, we can choose a different invertible submatrix and left-over column, say $\widehat{M}_1$ and $\widehat{b}_1$. By the continuity of $DH$, $\widehat{M}_1$ is invertible over some interval $s \in (s_{1/2}, s_2)$ with $0 < s_{1/2} < s_1 < s_2$. Using the same approach, we can construct $\mathcal{N}_1 \in ISDnet(m, n, 1)$ to compute the integral curve over $s \in (s_{1/2}, s_2)$. We can now $s$-glue $\mathcal{N}_0$ and $\mathcal{N}_1$ (Lemma 39) to construct $\mathcal{N}$ computing the integral curve across $s \in [0, s_2)$.

All that remains it to prove this procedure can be repeated until time $s = 1$ is reached. Due to the clamping in line 7 of Algorithm 1, $z$ remains in $[-c_{\max}, c_{\max}]^{n+1}$ for time $s \in [0, 1]$. Since $\mathcal{N} \in ISDnet(m, n, 1)$ is a piecewise polynomial, it is Lipschitz on the compact domain $\{x\} \times [-c_{\max}, c_{\max}]$, which means there exists $\epsilon > 0$ such that for each step $\tau$ of the $s$-glueing procedure above, we can choose $s_{\tau+1}$ such that $s_\tau + \epsilon < s_{\tau+1}$, and the conclusion follows. $\qquad\square$

### E.5 Discontinuous functions

We are now ready to directly tackle discontinuous semialgebraic functions. Our strategy is to split the computation into two distinct phases.

1. First, we utilize the fact that $\dot{z}$ can be discontinuous across a boundary where $M$ is singular to separate the domain $\mathbb{R}^m$ into the connected components of the graph of $F$. Every point in the $j$'th connected component is mapped to $(y_{\text{out}}, t_{\text{out}}) = (0, j)$.

2. Then, using the homotopy continuation arguments from Section 4.1, we construct continuous ISD phase vector fields that map from $(0, j)$ to $(F(x), t_{\text{out}})$, which was desired.

Furthermore, we carefully construct each piece of the computation to lie in disjoint $(t, s)$-space, so they can be glued together in a way consistent with a single continuous ISD network $\mathcal{N}$. This $\mathcal{N}$ computes $F$ while solving the ODE (6)–(7). Ensuring that the required $\mathcal{N}$ is indeed an ISD network (rather than an arbitrary continuous one) is one of the primary challenges in these constructions.

### E.6 CHARACTERISTIC FUNCTIONS

Given $S \subseteq \mathbb{R}^m$, a function $\chi_S : \mathbb{R}^m \to \mathbb{R}$ is the *characteristic function* of a set $S$ if $\chi_S(x) = 1$ for $x \in S$, and $\chi_S(x) = 0$ otherwise. We now show that SANNs can represent characteristic functions on semialgebraic sets.

**Lemma 41** (Scalar multiplication). *Given $\mathcal{N}$, $c_{\max}$, $y_0$, $t_0$, $s_0$, $s_{\text{final}}$ and $\alpha \in \mathbb{R}$, there exists $\mathcal{N}_1 \in ISDnet(m, n, 1)$ such that for all $x$,*

$$\texttt{ODESolve}(\mathcal{N}_1, c_1, x, (y_0, t_0, s_0), s_{\text{final}}) = \alpha \texttt{ODESolve}(\mathcal{N}, c_{\max}, x, (y_0, t_0, s_0), s_{\text{final}}) \quad (56)$$

*where $c_1 = \alpha c_{\max}$.*

*Proof.* When $\alpha = 0$, we simply use $M_1 = M$, $b_1 = 0$. Otherwise, we dilate and scale the phase vector field by $\alpha$, which we accomplish by setting

$$M_1(x, y, t, s) = M(x, \alpha^{-1}y, \alpha^{-1}t, s) \quad (57)$$

$$b_1(x, y, t, s) = \alpha b(x, \alpha^{-1}y, \alpha^{-1}t, s). \quad (58)$$

$\square$

**Lemma 42** (Changing $s$ bounds $s_{\text{final}}$). *Given $\mathcal{N}_1$, $c_1$, $y_0$, $t_0$, $s_0$, $s_0'$, $s_{\text{final}}$, $s_{\text{final}}'$, there exists $\mathcal{N}_2$ and $c_2$ such that for all $x$,*

$$\texttt{ODESolve}(\mathcal{N}_2, c_2, x, (y_0, t_0, s_0'), s_{\text{final}}') = \texttt{ODESolve}(\mathcal{N}_1, c_1, x, (y_0, t_0, s_0), s_{\text{final}}). \quad (59)$$

*Proof.* Let $\lambda(s) = (s_{\text{final}}' - s)/(s_{\text{final}}' - s_0')$ and $\alpha = (s_{\text{final}} - s_0)/(s_{\text{final}}' - s_0')$. Dilate, scale, and translate the phase vector field:

$$M_2(x, y, t, s) = M_1(x, y, t, s_0 + \lambda(s)(s_{\text{final}} - s_0)) \quad (60)$$

$$b_2(x, y, t, s) = \alpha b_1(x, y, t, s_0 + \lambda(s)(s_{\text{final}} - s_0)). \quad (61)$$

Set $c_2 = \alpha c_1$. $\square$

**Lemma 43** (Shifting initial conditions). *Given $\mathcal{N}_1$, $c_1$, $y_0$, $y_0'$, $t_0$, $t_0'$, $s_0$, and $s_{\text{final}}$, there exists $\mathcal{N}_2$ such that for all $x$,*

$$\texttt{ODESolve}(\mathcal{N}_2, c_1, x, (y_0', t_0', s_0), s_{\text{final}}) = (y_0', t_0') + \texttt{ODESolve}(\mathcal{N}_1, c_1, x, (y_0, t_0, s_0), s_{\text{final}}). \quad (62)$$

*Proof.* Let $\delta y = y_0' - y_0$ and $\delta t = t_0' - t_0$.

$$M_2(x, y, t, s) = M_1(x, y - \delta y, t - \delta t, s) \quad (63)$$

$$b_2(x, y, t, s) = b_1(x, y - \delta y, t - \delta t, s). \quad (64)$$

$\square$

The next lemma allows for a limited form of addition in the $t$ output.

**Lemma 44** (Addition in $t$). *Given $\mathcal{N}_1$, $\mathcal{N}_2$, $c_1$, $c_2$, $y_0$, $t_0$, $s_0$, $s_{\text{final}}$, suppose the image of*

$$x \mapsto \Pi_{n+1} \texttt{ODESolve}(\mathcal{N}_1, c_1, x, (y_0, t_0, s_0), s_{\text{final}}) \tag{65}$$

*has only finitely many points $(t_{out}^1, \ldots, t_{out}^J)$. Further suppose*

$$\text{diam}\{\Pi_{n+1}\texttt{ODESolve}(\mathcal{N}_2, c_2, x, (y_0, t_0, s_0), s) \mid s \in (s_0, s_{\text{final}})\} < \frac{1}{2} \min_{i,j} |t_{out}^i - t_{out}^j|. \tag{66}$$

*Then there exists $\mathcal{N}_3$, $c_3$ such that*

$$\Pi_{n+1}\texttt{ODESolve}(\mathcal{N}_3, c_3, x, (y_0, t_0, s_0), s_{\text{final}}) = \tag{67}$$
$$\Pi_{n+1}\left(\texttt{ODESolve}(\mathcal{N}_1, c_1, x, (y_0, t_0, s_0), s_{\text{final}}) + \texttt{ODESolve}(\mathcal{N}_2, c_2, x, (y_0, t_0, s_0), s_{\text{final}})\right). \tag{68}$$

*Proof.* Let $s_{1/2} = s_0 + (s_{\text{final}} - s_0)/2$. From Lemma 42, there exists $\mathcal{N}_1'$ and $c_1'$ such that

$$\texttt{ODESolve}(\mathcal{N}_1', c_1', x, (y_0, t_0, s_0), s_{1/2}) = \texttt{ODESolve}(\mathcal{N}_1, c_1, x, (y_0, t_0, s_0), s_{\text{final}}). \tag{69}$$

Likewise, from Lemmas 42 and 43, for each $j = 1, \ldots, J$, there exists $\mathcal{N}_2^j$ and $c_2^j$ such that

$$\texttt{ODESolve}(\mathcal{N}_2^j, c_2^j, x, (y_{out}^j, t_{out}^j, s_{1/2}), s_{\text{final}}) = \tag{70}$$
$$(y_{out}^j, t_{out}^j) + \texttt{ODESolve}(\mathcal{N}_2, c_2, x, (y_0, t_0, s_0), s_{\text{final}}). \tag{71}$$

Hypothesis 66 guarantees the sets

$$\{\Pi_{n+1}\texttt{ODESolve}(\mathcal{N}_2^j, c_2^j, x, (y_{out}^j, t_{out}^j, s_{1/2}), s) \mid s \in [s_{1/2}, s_{\text{final}}]\} \tag{72}$$

are pairwise disjoint for each $j$, so the $\mathcal{N}_2^j$ can be $t$-glued (Lemma 40) into a single $\mathcal{N}_3'$ such that

$$\texttt{ODESolve}(\mathcal{N}_3', c_3', x, (y_{out}^j, t_{out}^j, s_{1/2}), s_{\text{final}}) = (y_{out}^j, t_{out}^j) + \texttt{ODESolve}(\mathcal{N}_2, c_2, x, (y_0, t_0, s_0), s_{\text{final}}) \tag{73}$$

for $j = 1, \ldots, J$. Finally, using Lemma 39, $s$-glue $\mathcal{N}_1'$ and $\mathcal{N}_3'$ at $s = s_{1/2}$ to obtain the desired $\mathcal{N}_3$. $\quad\square$

We are now ready to construct SANNs that represent characteristic functions of arbitrary semialgebraic sets.

**Proposition 45.** *Let $S \subset \mathbb{R}^m$ be a semialgebraic set. Then there exists $\mathcal{N}$ and $C \in \mathbb{N}$ such that for all $c \geq C$,*

$$\texttt{ODESolve}(\mathcal{N}, c, x, (0, 0, 0), 1) = (0, \chi_S(x)). \tag{74}$$

*Proof.* First, suppose $S$ is open. Then $S^c$ is a closed semialgebraic set, and from Proposition 7 we conclude there exists $g \in ISD(\mathbb{R}^m, \mathbb{R}_{\geq 0})$ such that $\ker(g) = S^c$. Define

$$M(x, y, t, s) = g(x)I \tag{75}$$
$$b(x, y, t, s) = g(x)(0, 1). \tag{76}$$

$M$ is singular iff $x \in \ker(g)$, so $\dot{z}$ always returns 0 in this case. Otherwise, $(\dot{y}, \dot{t}) = M^{-1}b = (0, 1)$. Integrating from time $s_0 = 0$ to $s_{\text{final}} = 1$ yields

$$\texttt{ODESolve}(\mathcal{N}, c, x, (0, 0, 0), 1) = \begin{cases} (0, 1), & x \notin \ker g \\ (0, 0), & \text{otherwise,} \end{cases} \tag{77}$$

which proves the proposition for open $S$.

For closed $S$, recall $S^c$ is open and $\chi_S = 1 - \chi_S^c$, so Lemmas 41 and 44 yield the result.

Next, assume we can represent the characteristic function for $S_1$ and $S_2$. We will show how to represent the characteristic function of their intersection; equivalently, we show how to represent $\chi_{S_1 \cap S_2^c}$. Let $\mathcal{N}_j$ be such that

$$\texttt{ODESolve}(\mathcal{N}_j, c, x, (0, 0, 0), 1) = (0, \chi_{S_j}(x)) \tag{78}$$

for $j = 1, 2$. Using Lemma 42, there exists $\mathcal{N}_1'$ such that

$$\texttt{ODESolve}(\mathcal{N}_1', c, x, (0,0,0), 1/2) = (0, \chi_{S_1}(x)). \tag{79}$$

Using Lemmas 42, 43, and 41, there exists $\mathcal{N}_2'$ such that

$$\texttt{ODESolve}(\mathcal{N}_2', c, x, (0, 1, 1/2), 1) = (0, 1) - (0, \chi_{S_2}(x)). \tag{80}$$

Now $t$-glue $\mathcal{N}_2'$ with a trivial ISD network so that

$$\texttt{ODESolve}(\mathcal{N}_2', c, x, (0, 0, 1/2), 1) = (0, 0), \tag{81}$$

and $s$-glue $\mathcal{N}_1'$ with $\mathcal{N}_2'$ at $s = 1/2$ to obtain the desired network.

We can now handle the case of an arbitrary semialgebraic set $S$. The key is to exploit the "cylindrical decomposition" of $S$: every semialgebraic set in $\mathbb{R}^m$ is the union of finitely many disjoint sets homeomorphic to $(-1, 1)^d$, $d \leq m$ (Bochnak et al., 1998). In particular, it is the union of finitely many disjoint locally closed sets. Since locally closed sets are the intersection of an open and a closed set, we have already shown how to represent their characteristic functions with SANNs. Furthermore, since these locally compact sets are disjoint, the characteristic function of their union can be represented by summing the individual characteristic functions with $s$-glueing. $\qquad\square$

*Remark.* Note that, although $M = g(x)I$ is in some sense "nearly singular" close to the kernel of $g$, it is still in fact an orthogonal matrix. No numerical instabilities arise from the conditioning of $M$ when solving $M\dot{z} = b$.

### E.7 PROOF OF THEOREM 13

**Theorem 13.** *Let $F : \mathbb{R}^m \to \mathbb{R}^n$ be a (possibly discontinuous) bounded semialgebraic function. Then there exists $\mathcal{N} \in ISDnet(m, n, 1)$ and $c_{\max} \in \mathbb{R}_{\geq 0}$ such that $SANN^{lim}(\mathcal{N}, \cdot, c_{\max}) = F$.*

*Proof.* Every semialgebraic function has finitely many connected components (Bochnak et al., 1998), so let $\{S_j\}_{j=1}^K$ be a semialgebraic decomposition of $\mathbb{R}^m$ such that $F$ is continuous on each $S_i$.

Using Proposition 45, we can construct scaled characteristic functions on each connected component, and using Lemmas 41 and 44, we scale and add these functions to obtain $\mathcal{N}_0, c_0$ such that

$$\texttt{ODESolve}(\mathcal{N}_0, c_0, x, (0, 0, 0), 1/2) = (0, 3j) \text{ where } x \in S_j. \tag{82}$$

Using Theorem 12, let $\mathcal{N}_j, c_j$ be such that

$$SANN(\mathcal{N}_j, \cdot, c_j)|_{S_j} = F|_{S_j} \tag{83}$$

for each $j = 1, \ldots, K$. Furthermore, by the construction in the proof of that theorem, $\mathcal{N}_i$ can be chosen such that $t(0) = 0$ and $t(1) = 1$. In particular,

$$\texttt{ODESolve}(\mathcal{N}_j, c_j, x, (0, 0, 0), 1) = (F|_{S_j}(x), 1) \text{ when } x \in S_j. \tag{84}$$

Using Lemma 42, modify each $\mathcal{N}_i, c_i$, so that integration occurs in the interval $s \in [1/2, 1]$. Furthermore, using Lemma 43, modify each $\mathcal{N}_i, c_i$ to begin integration at $(0, 3j)$. Finally, $t$-glue these modified $\mathcal{N}_j$, and $s$-glue the result to $\mathcal{N}_0$ at $s = 1/2$. $\qquad\square$

### E.8 SEMIALGEBRAIC FUNCTIONS WITH UNBOUNDED IMAGE

Parameter $c_{\max}$ provides an upper bound on the $\ell_\infty$-norm of the output of SANN, so the image of every SANN-representable function is bounded. However, the same is not true for all semialgebraic functions, even those with compact domain, such as $F(x) : [-1, 1] \to \mathbb{R}$ defined by

$$F(x) = \begin{cases} 1/x, & x \neq 0, \\ 0 & x = 0. \end{cases} \tag{85}$$

Such functions can not be represented by SANNs.

### E.9 SANNs ALWAYS COMPUTE SEMIALGEBRAIC FUNCTIONS

In our considerations, we use the Tarski–Seidenberg theorem, which has several equivalent formulations. We use the following:

**Theorem 46** (Tarski–Seidenberg). *When $X \subset \mathbb{R}^n \times \mathbb{R}^k$ is a semialgebraic set and $\pi : \mathbb{R}^n \times \mathbb{R}^k \to \mathbb{R}^n$ is the projection map $\pi(x', x'') = x'$, then the set $\pi(X)$ is a semialgebraic set in $\mathbb{R}^n$.*

So far, we have shown that SANNs are able to represent any bounded semialgebraic function. For completeness, we now prove a partial converse; that is, SANNs always compute semialgebraic functions provided the numerical ODE solver is semialgebraic. Common ODE solvers, such as Euler and Runge-Kutta methods, are semialgebraic. The proposition below focuses specifically on the case of Forward Euler timestepper given in Section 3.2, but it can be easily modified for other methods.

**Proposition 47.** *For all $\mathcal{N} \in ISDnet(m, n, k)$, $x \in \mathbb{R}^m$, and $c_{max} > 0$, the function $f_{\mathcal{N}, c_{max}} : x \mapsto \Pi z_N$, where $z_N$ is given by the iteration (2), is semialgebraic.*

*Proof.* The operation iteration steps $(x, j, z_j) \mapsto z_{j+1}$ are a composition of ISD-function and the function $(M, b) \mapsto (\det(M), M_c, b)$, where $M_c$ is the co-factor matrix of $M$, the function

$$(\det(M), M_c, b) \mapsto (\mathbf{1}_{\det(M)>0} + \mathbf{1}_{\det(M)<0}) (\det(M))^{-1} M_c b$$

that implements Cramer's rule, and another ISD-function given by the time-step formula

$$(z_j, \dot{z}_j) \to z_j + \frac{1}{N} \dot{z}_j.$$

This makes the function $x \mapsto z_N$ semialgebraic; including the projection $\Pi_n$ results in a semialgebraic function due to the Tarski–Seidenberg theorem. $\qquad\square$

## F NUMERICAL EXAMPLE: INVERSE PROBLEM FOR ELECTRICAL RESISTOR NETWORKS

This section demonstrates the feasibility of training SANNs to solve a difficult non-linear inverse problem.

Suppose we are given an electrical network $\Gamma = (V, E)$ with vertices $V$ and edges $E \subset V \times V$. Physically, an electrical network is a system of connected resistors, and it is modeled by a graph where every edge is a resistor. An inverse problem for an electrical network is to find one or all graphs and resistors on the edges when the outcome of all possible voltage and current measurements in some of vertices of the graph (called the boundary nodes) are given, see Figure 5.

We partition $V$ into boundary nodes $V_B$ and interior nodes $V_I$. Let $\gamma : E \to \mathbb{R}_{\geq 0}$ be the edge *conductivities*. The *network response matrix* (i.e. Dirichlet-to-Neumann map) $\Lambda \in \mathbb{R}^{|V_B| \times |V_B|}$ computes the current at the boundary nodes resulting from applying voltages on $V_B$. Our task is to recover the conductivites $\gamma$ from measurements $\Lambda$ of voltage-response on the boundary. See Curtis & Morrow (2000) for a thorough treatment of this inverse problem, and Forcey (2023) for recent developments.

Equivalently, we wish to recover the *Kirchhoff matrix* $K \in \mathbb{R}^{|V| \times |V|}$ where

$$K_{ij} = \begin{cases} \gamma(i, j) & \text{if } (i, j) \in E \\ -\sum_{k \neq i} \gamma(i, k) & \text{if } i = j \\ 0 & \text{otherwise.} \end{cases} \tag{86}$$

$K$ is a symmetric matrix with conductivities on off-diagonals and that satisfies Kirchhoff's Law—the sum of entries in each row is 0. Curtis & Morrow (2000) showed that $\Lambda$ is the Schur complement of a particular submatrix of the Kirchhoff matrix. Precisely, we write

$$K = \begin{bmatrix} A & B \\ B^T & D \end{bmatrix} \tag{87}$$

where $A$ corresponds to boundary nodes $V_B$ and $D$ corresponds to interior nodes $I$. If we let $\mathcal{K}(\Gamma) \subset \mathbb{R}^{|V| \times |V|}$ denote the set of valid Kirchhoff matrices on $\Gamma$ (a semialgebraic set), then the forward map (mapping a graph and the values of the resistors to the values of the measurements) is the semialgebraic function $F : \mathcal{K}(\Gamma) \to \mathbb{R}^{|V_B| \times |V_B|}$ defined

$$F(K) := A - BD^{-1}B^T = \Lambda \tag{88}$$

The inverse problem is to recover an element of the preimage $F^{-1}(\{\Lambda\})$. Since preimages of semialgebraic sets (in this case the single point $\{\Lambda\}$) of semialgebraic functions ($F$) are themselves semialgebraic (Bochnak et al., 1998), we conclude $F^{-1}(\{\Lambda\})$ is a semialgebraic set.

In certain special network topologies, such as for rectangular networks (Figure 5), the preimage of $F^{-1}(\{\Lambda\})$ is a singleton, that is, the values $\gamma(i, j)$, $(i, j) \in E$ of the resistors on the graph $(V, E)$ are uniquely determined by the voltage-to-current map $\Lambda$ (Curtis & Morrow, 2000).

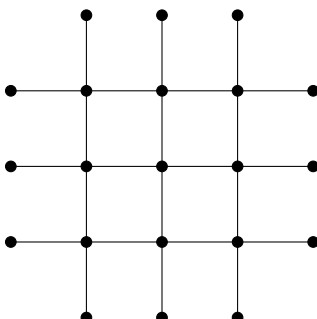

Figure 5: A $3 \times 3$ rectangular electrical network. Each edge in the graph represents a resistor with unknown conductance.

This inverse problem has had a wide impact in several fields. In medical imaging it has led to new imaging modalities, for example, in monitoring lung function. Recently, EIT is used in monitoring the lungs of the COVID-19 patients, see Cappellini et al. (2024).

### F.1 NUMERICAL RESULTS FROM TRAINING.

We demonstrate the feasibility of training SANNs to perform non-trivial tasks using the network EIT problem as an example. Specifically, given a voltage-response map $\Lambda$ for a $2 \times 2$ rectangular electrical network, the task is to recover the non-zero, off-diagonal entries of the Kirchhoff matrix $K$. We trained a 4-layer SANN to minimize the loss $\mathcal{L}_{\text{total}}$ from equation (9). We compare this against a feed-forward neural network trained to minimize $\mathcal{L}_{\text{accuracy}}$ from equation (9) on the same data. Both networks were trained using the Adam optimizer. We also compare against a known reconstruction algorithm from Curtis & Morrow (2000) that is exact for noiseless data. The results are shown in Table 2 below. Let us shortly discuss these results: The Curtis and Morrow (C&M) algorithm is an analytical method that solves the problem using the theory of planar graphs, and therefore it is provably correct for all inputs, including inputs outside the training data used for the other algorithms, such as exceptional resistor networks with very small or large resistors. However, the algorithm is very sensitive to noise. The algorithm based on a feed forward network is used in our tests as a black-box algorithm and it works well with noisy data. The SANN is between these two algorithms—it is explainable in the sense that SANN can represent the rational matrix functions, and it works moderately well with noisy data. We note that in the numerical tests the training of SANNs was not fine tuned and by improving the training methods, it is possible that the results could be improved.

As in the linear inverse problem example from Section 5.2, a trained SANN performs comparably to a standard feed-forward neural network of the same size, demonstrating the feasibility of training these networks. We do not claim to demonstrate numerical advantages of SANNs at this stage; further research is necessary to develop new techniques that fully leverage their expressive power.

Both networks outperform the C&M algorithm on noisy input data. The C&M algorithm reconstructs conductances edge-by-edge, starting from the boundary and working inward; this causes

| | # parameters | Relative error (noiseless) | Relative error (1% noise)) |
|---|---|---|---|
| Feed forward network | $17,712$ | $0.0157$ | **0.0180** |
| SANN | $17,640$ | $0.0234$ | $0.0273$ |
| C&M | — | **2.49e − 07** | $0.0469$ |

Table 2: Results of training neural networks from data to solve the network EIT problem on $2 \times 2$ rectangular electrical networks. The last row (C&M) shows the reconstruction algorithm from Curtis & Morrow (2000).

errors to accumulate rapidly for interior edges. In contrast, neural networks incorporate learned prior knowledge of the expected distribution of conductances, making them more robust to noisy measurements within the data distribution.

### F.2 CONTINUOUS EIT PROBLEM (PDEs)

The inverse problem for electrical networks described above is a discrete version of the medical imaging problem encountered in Electrical Impedance Tomography (Borcea, 2002), an inverse problem for finite element models for the (anisotropic) conductivity equation. If $\Omega \subset \mathbb{R}^2$ is the interior of a simple polygon, and if one is given a triangulation of $\overline{\Omega}$, then the Finite Element Method (FEM) model for the Dirichlet problem for an elliptic partial differential equation,

$$\nabla \cdot (\sigma(x) \nabla u(x)) = 0, \quad x \in \Omega, \quad u|_{\partial \Omega} = F, \tag{89}$$

for a given matrix or scalar valued conductivity function $\sigma(x)$, is the matrix equation

$$\begin{pmatrix} I_B & 0 \\ L' & L'' \end{pmatrix} u = \begin{pmatrix} f \\ 0 \end{pmatrix}$$

where $u = (u_1 \cdots u_N)^t$ corresponds to values of the FEM solution at the vertices of the triangulation, $f = (f_1 \cdots f_B)^t$ corresponds to the boundary value $f$ at the boundary vertices, and the matrix $(L'\ L'')$ depends on $\sigma$ and the triangulation and has rows whose elements add up to zero.

If one considers the triangulation of $\overline{\Omega}$ as a graph, and if to each edge of the graph one assigns a resistor with a given conductivity, then the Dirichlet problem for this resistor network is to find a solution $u = (u_1 \cdots u_N)^t$, corresponding to voltages at each vertex, of the matrix equation

$$\begin{pmatrix} I_B & 0 \\ K' & K'' \end{pmatrix} u = \begin{pmatrix} f \\ 0 \end{pmatrix} \tag{90}$$

where $f = (f_1 \cdots f_B)^t$ corresponds to voltages at the boundary vertices. Also here, the rows of the matrix $(K'\ K'')$, which depends on the resistors, add up to zero.

Thus, formally, the FEM model for an elliptic partial differential equation and resistor networks lead to the same kind of equation, see Lassas et al. (2015) for details.

Electrical Impedance Tomography (EIT) is a method proposed for medical and industrial imaging, where the objective is to determine the electrical conductivity or impedance in a medium by making voltage to current measurements on its boundary (Borcea, 2002). It has led to new imaging modalities, for example the Pulmovista500 device by Draeger AG & Co. used to monitor lung function. Recently, EIT is being used to monitor the lungs of COVID-19 patients (Cappellini et al., 2024).

The boundary measurements are modeled by the Dirichlet-to-Neumann map $\Lambda_a$, which maps the boundary value $f$ of the electric voltage to the electric current through the boundary (i.e., the conormal derivative) $\Lambda_a(f) = \nu \cdot \sigma u|_{\partial \Omega}$.

In FEM, let $\Omega$ be the interior of a simple polygon, and fix a triangulation of $\overline{\Omega}$ with vertices $\{x_1, \ldots, x_N\}$ of which $\{x_1, \ldots, x_B\}$ lie on $\partial \Omega$. Let $A(\Omega)$ be the set of continuous functions $\overline{\Omega} \to \mathbb{R}$ which are affine on each triangle. If $v_k$ is the unique function in $A(\Omega)$ which is 1 at $x_k$ and 0 at all the other vertices, then $A(\Omega) = \{\sum_{k=1}^{N} c_k v_k \, ; \, c_k \in \mathbb{C}\}$.

There is a voltage-to-current map $\Lambda_\sigma^{\mathrm{FEM}}$ related to the FEM model that is a discrete version of the Dirichlet-to-Neumann map. To define this map, let us consider the space $A(\partial \Omega) = A(\Omega)|_{\partial \Omega} =$

$\{\sum_{j=1}^{B} c_j v_j|_{\partial\Omega}; c_j \in \mathbb{R}\}$, and a quadratic form $Q_\sigma$ on $A(\partial\Omega)$ that is given for $f \in A(\partial\Omega)$ by

$$Q_\sigma(f, f) = \min_u \int_\Omega \sigma(x) \nabla u(x) \cdot \nabla u(x) \, dx,$$

where the minimum is taken over $u \in A(\Omega)$ satisfying the boundary condition $u|_{\partial\Omega} = f$. The voltage-to-current map $\Lambda_\sigma^{\text{FEM}}$ is a symmetric matrix $\Lambda_\sigma^{\text{FEM}} : \mathbb{R}^B \to \mathbb{R}^B$ determined by the formula

$$\vec{f} \cdot \Lambda_\sigma^{\text{FEM}} \vec{f} = Q_\sigma(f, f)$$

where $f = \sum_{j=1}^{B} f_j v_j|_{\partial\Omega}$ is the boundary current corresponding to a vector $\vec{f} = (f_j)_{j=1}^{B}$.

The solution of the matrix equation (90) can be found using Cramer's formula as a meromorphic function. This implies that there is a SANN that solves the FEM approximation problem (90) for the partial differential equation (89) and another SANN that represents the voltage-to-current map.

# G ADDITIONAL APPLICATIONS

## G.1 SEMIALGEBRAIC OPTIMIZATION

Consider the general class of optimization problems

$$f(\theta) = \operatorname*{arginf}_{\theta^*} \quad g(\theta^*, \theta) \quad \text{subject to the condition } \theta^* \in \mathcal{S}(\theta) \subset \mathbb{R}^n \tag{91}$$

where $\theta \in \mathbb{R}^m$, and both $g : \mathbb{R}^n \times \mathbb{R}^m \to \mathbb{R}$ and $\mathcal{S} : \mathbb{R}^m \to \mathcal{P}(\mathbb{R}^n)$ are semialgebraic functions. Here, $\mathcal{P}(\mathbb{R}^n) := \{x \mid x \subset \mathbb{R}^n\}$ is the power set of $\mathbb{R}^n$, and $\mathcal{S}(\theta)$ is a semialgebraic constraint set that is allowed to vary based on $\theta$. Using the Tarski–Seidenberg theorem, we can show that $f$ is a (possibly set-valued) semialgebraic function. Indeed, the graph of $f$ is

$$\left\{ \left( \theta, \Pi_n \partial \left( \overline{\operatorname{epi}(g)} \cap (\mathcal{S}(\theta) \times \mathbb{R}^m \times \mathbb{R}) \right) \right) \mid \theta \in \mathbb{R}^m \right\} \tag{92}$$

where $\Pi_n$ projects onto the first $n$ coordinates, $\partial$ is the boundary operation, and $\overline{\operatorname{epi}(g)}$ is the closure of the epigraph $\operatorname{epi}(g) = \{(\theta^*, \theta, t) \in (\mathbb{R}^n \times \mathbb{R}^m \times \mathbb{R}) \mid t \geq g(\theta^*, \theta)\}$ of $g$. Since this is a semialgebraic set, $f$ is a semialgebraic function. If we impose the additional assumption that $f$ is bounded (which holds if $\mathcal{S}$ is bounded, for example), then by Theorem 13 it can be represented by a SANN.

*Remark.* Optimization problem (91) includes as special cases polynomial optimization and quadratic/linear/mixed-integer programming. It is at least NP-hard; there is no general polynomial-time algorithm to compute $f$ from a description of $g$ and $\mathcal{S}$ (Lasserre, 2015).

Many interesting problems can be formulated in terms of (91). Below, we give a new potential application using SANNs as a hypernetwork to sparsify ReLU networks.

## G.2 SANNS AS HYPERNETWORKS

In this section, we demonstrate how SANNs can be used as hypernetworks to optimize various properties of target ReLU networks.

To connect the general class of optimization problems (91) to hypernetworks, suppose we have a fixed target architecture $\mathcal{N}_{\text{target}}$ with trained weights $\theta \in \mathbb{R}^m$. Let $\mathcal{S}(\theta) \subset \mathbb{R}^m$ be the set of all weights generating *equivalent* networks (Petersen et al., 2021); i.e. every choice of weights from $\mathcal{S}(\theta)$ produces the same input-output pairs as $\theta$ when used in architecture $\mathcal{N}_{\text{target}}$. When $\mathcal{N}_{\text{target}}$ is a feed-forward ReLU network, $\mathcal{S}(\theta)$ is a semialgebraic set (Valluri & Campbell, 2023).

Now, different choices of objective function $g$ optimize various properties of the target network. For example, when $g$ counts the number of non-zero elements of $\theta^*$ (the "$L^0$ norm"), $f$ returns the sparsest-possible equivalent network. Using sparse $\theta^*$ is both more memory-efficient (Bölcskei et al., 2018) and interpretable (Fan et al., 2021) than using $\theta$. Likewise, $f$ can return a "nearly-equivalent" network with small Lipschitz constant by relaxing $\mathcal{S}$ and using $g$ that computes the Lipschitz constant of $\mathcal{N}_{\text{target}}$ with weights $\theta^*$. Reducing the Lipschitz constant of the target network improves its robustness, interpretability, and generalization performance (see Ducotterd et al. (2024) and references therein).

### G.3 SEMIALGEBRAIC TRANSFORMER

In this section, we demonstrate how a variation of the transformer architecture (Vaswani et al., 2023) is expressible as a SANN. Our presentation of the transformer architecture is inspired by Furuya et al. (2024).

We start with the SoftMax attention mechanism at the heart of the transformer architecture. $\text{SoftMax} : \mathbb{R}^{n \times n} \to \mathbb{R}_+^{n \times n}$ operates row-wise on its input matrix:

$$\text{SoftMax}(X)_{i,j} := \frac{\exp(X_{i,j})}{\sum_{\ell=1}^n \exp(X_{i,\ell})}. \tag{93}$$

SoftMax is a continuous approximation to the row-wise function $\arg\max : \mathbb{R}^n \to \mathbb{R}^n$

$$(\arg\max x)_i := \begin{cases} 1/s & \text{if } x_i \geq x_j \text{ for all } j = 1 \ldots n, \\ 0 & \text{otherwise.} \end{cases} \tag{94}$$

where $s$ is the number of indices $j$ such that $x_j = \max x$. It is not a semialgebraic function since $\exp : \mathbb{R} \to \mathbb{R}$ is not semialgebraic; however, for our purposes we can use the following piecewise-rational function $\text{saxp} : \mathbb{R} \to \mathbb{R}$ instead:

$$\text{saxp}(x) = \begin{cases} x^2 + x + 1 & x \geq 0 \\ 1/(x^2 - x + 1) & x < 0. \end{cases} \tag{95}$$

saxp shares the following properties with $\exp$:

- $\text{Im}(\text{saxp}) = \text{Im}(\exp) = (0, +\infty)$
- $\text{saxp}(0) = \exp(0)$ and $\text{saxp}'(0) = \exp'(0)$
- $\lim_{x \to -\infty} \text{saxp}(x) = 0$, and $\lim_{x \to +\infty} \text{saxp}(x) = +\infty$.

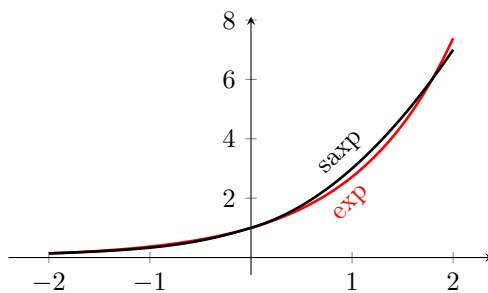

Figure 6: saxp (black) is a semialgebraic approximation to $\exp$ (red). It is simple, accurate near $0$, and has the correct asymptotic behavior as $x \to \pm\infty$.

saxp is a good approximation of $\exp$ near $0$, but $\text{saxp} = o(\exp)$ as $x \to \infty$. In fact this is true for every semialgebraic function since every semialgebraic function is bounded above by a polynomial (Bochnak & Efroymson, 1980). This is not a concern for our transformers since layer normalization will keep the input to saxp near $0$.

By using saxp in place of $\exp$ in equation (93), we define SArgMax ("Semialgebraic ArgMax"):

$$\text{SArgMax}(X)_{i,j} := \frac{\text{saxp}(X_{i,j})}{\sum_{\ell=1}^n \text{saxp}(X_{i,\ell})}. \tag{96}$$

Like SoftMax, every entry of $\text{SArgMax}(X)$ lies in the interval $(0, 1)$, and every row sums to $1$. This yields the semialgebraic attention mechanism

$$\text{Att}_{\theta^h}(X) := VX \, \text{SArgMax}(X^T Q^T K X / \sqrt{k}) \tag{97}$$

where $\theta^h := (K, Q, V) \in \mathbb{R}^{k \times d_{\text{in}}} \times \mathbb{R}^{k \times d_{\text{in}}} \times \mathbb{R}^{d_{\text{head}} \times d_{\text{in}}}$ ("Key", "Query", and "Value") are learnable parameters. A multi-head attention mechanism with a skip connection is simply a linear combination of attention blocks:

$$\text{MAtt}_\theta(X) := X + \sum_{h=1}^H W^h \text{Att}_{\theta^h}(X) \tag{98}$$

where $\theta := (W^h, \theta^h), h = 1, \ldots H$ are the learnable parameters specific to each attention head.

Transformers are constructed by composing multi-head attention, MLP, and layer normalization blocks. MLP blocks with ReLU activation are simply

$$\text{MLP}_\theta(X) := \text{ReLU}(WX + B) \tag{99}$$

with parameters $\theta := (W, B)$. Row-wise layer normalization is

$$\text{LayerNorm}(X)_{i,j} := \frac{X_{i,j} - \mu_i}{\sigma_i} \qquad \text{where} \tag{100}$$

$$\mu_i := \frac{1}{n} \sum_{\ell=1}^{n} X_{i,\ell} \tag{101}$$

$$\sigma_i := \sqrt{\frac{1}{n} \sum_{\ell=1}^{n} (X_{i,\ell} - \mu_i)^2}. \tag{102}$$

Recall that every function defined piecewise on a semialgebraic decomposition of $\mathbb{R}^m$ using only addition, subtraction, multiplication, division, and extraction of roots is a semialgebraic function. From this, it is clear that $\text{MAtt}$, $\text{MLP}$, and $\text{LayerNorm}$ in equations (98), (99), and (100) are semialgebraic. Furthermore, they are all continuous, so they are bounded when their domain is restricted to be compact, such as after a tokenization step. Finally, note that the composition of semialgebraic functions is itself a semialgebraic function.

Transformers with semialgebraic attention are built as compositions of equations (98), (99), and (100), so they are bounded semialgebraic functions and thus representable by a SANN by Theorem 13.

We emphasize that our only change to the transformer was to replace $\exp$ with the semialgebraic approximation $\text{saxp}$. This modified transformer is naturally expressible as a SANN without further modification of our architecture.

