# OpenReview forum: "Semialgebraic Neural Networks: From roots to representations"
_ICLR.cc/2025/Conference — ICLR 2025 Poster_

### Official Review · Reviewer_3yC9 · 2024-10-28

**Soundness:** 2
**Presentation:** 2
**Contribution:** 3
**Rating:** 5
**Confidence:** 3

**Summary:**

This paper introduces Semialgebraic Neural Networks (SANNs), a novel neural architecture designed to exactly compute bounded semialgebraic functions through homotopy continuation methods. The key idea is to encode the graph of a learned function as the kernel of a piecewise polynomial and then use numerical ODE solvers to evaluate it. The authors provide theoretical guarantees for the expressivity of SANNs, showing they can represent both continuous and discontinuous bounded semialgebraic functions. The work bridges classical numerical analysis techniques with modern neural network architectures.

**Strengths:**

Overall interesting application of homotopy methods to learn semi-algebraic functions. The work includes nice background on semi-algebraic geometry. Overall the high level idea is potentially interesting.

**Weaknesses:**

The biggest weakness of the paper (and major reason for the score) is the lack of clarity in the definition of SANN's and Section 4. I mention some suggestions in the questions below.
The introduction hints at a potential answer, but I don't understand why one would want to use this framework. Is this better than using the traditional learning framework (of learning F directly?)
Here is a list of other weaknesses:
There is no implementation or numerical work. (This isn't necessary, but I think the claim that you "demonstrate on applications" isn't accurate.)
It isn't clear which results are classical, or almost classical with a few notational differences. For example, is Proposition 7 a new result? What is new in Section 4.2?

**Questions:**

Questions:
The questions are in no particular order.
1. Fig 1 caption: Why does homotopy not give parts in kernel of G that are not part of F but somehow also correctly gives isolated point (0,0)?
2. What does "defined in pieces" mean?
3. In corollary 8, "gr" is not defined.
4. Why not learn the kernel directly? (As in, learn the predictive model F(x)= y as is standard in ML) What is the benefit of learning G?
5. Why do we expect (7) to be small when s < 1?
6. Above Thm 16 you say "exactly compute" but in remark earlier you mentioned that this will still have discretization error due to ODE solve. Should this be changed?
7. Why does Theorem 14 only succeed with "probability 1"?

Suggestions:
1. Define acronym SANNS in main text.
2. Clarity of Definition 1 can be improved. Perhaps the free lattice should be defined? C^k(D, R^n) is not defined (I'm assuming this is just k times differentiable functions on R^n?)
**3. I do not understand the definition of SANNs in equations 2-5. How does little n affect Z(N) and what is the product over? What is M and what does the the tuple equation (M, b) = N mean? (I'm guessing but can't be certain it's like the clamp-sol argument.) I suggest clarifying the notation here more carefully.
4. The distinction between existing work and new contributions is not clear in 4.2. Can you write your results as a proposition?
5. Does Section 4.1 really belong in Section 2?
6. In your summary of Section 4, perhaps include the subsection numbers to help navigate the relationships between the proofs.

---

> ### Author Response · Authors · 2024-11-28
> **Weaknesses**
>
> We thank the reviewer for their thorough and insightful review of our submission. Their feedback has been invaluable in improving the clarity and quality of our manuscript. We hope the revised version meets their expectations and addresses their concerns. Regardless of the outcome, we greatly appreciate their contribution to strengthening our work.
>
> Please refer to our general overview of the major changes made in this revision. Below, we provide detailed responses to specific points raised by the reviewer.
>
> **Weaknesses**
>
> > The biggest weakness of the paper (and major reason for the score) is the lack of clarity in the definition of SANN's and Section 4.
>
> Also:
>
> > I do not understand the definition of SANNs in equations 2-5. How does little n affect Z(N) and what is the product over? What is M and what does the tuple equation $(M, b) = N$ mean? (I'm guessing but can't be certain it's like the clamp-sol argument.) I suggest clarifying the notation here more carefully.
>
> Thank you for pointing out these issues. We have extensively revised Section 3, where SANNs are defined, and moved much of the technical content from Section 4 to the appendices. Our aim in the new revision is to make the architecture clear and emphasize the intuition behind the mathematical representation theorems. We have also added an architecture diagram to aid understanding.
>
> ---
>
> > I mention some suggestions in the questions below. The introduction hints at a potential answer, but I don't understand why one would want to use this framework. Is this better than using the traditional learning framework (of learning F directly?)
>
> Also:
>
> > Why not learn the kernel directly? (As in, learn the predictive model F(x)= y as is standard in ML) What is the benefit of learning G?
>
> It is not clear if it is possible to directly parameterize an arbitrary semialgebraic function $F$ in a way amenable to machine learning. All we know about $F(x) = y$ is its graph can be described by finitely many polynomial equalities $p(x, y) = 0$ and inequalities $q(x, y) \ge 0$.
> Small changes to the coefficients of these polynomials could result in a set $S$ that is no longer the graph of a function—for example, there may be no $y$ such that $(x_0,y) \in S$ for some $x_0$​, meaning the "function" would no longer be defined at $x_0$​.
>
> To address this issue, we encode $F$ as the kernel of $G$, then design a network architecture capable of executing a globally convergent root finding procedure (specifically, a homotopy continuation method). In particular, this ensures that for any choice of network parameters for $\mathcal{N}$, the SANN produces an output.
>
> ---
>
> > Here is a list of other weaknesses: There is no implementation or numerical work. (This isn't necessary, but I think the claim that you "demonstrate on applications" isn't accurate.)
>
> In the revised version, we include numerical results in Section 5 and Appendix F. We have also made a clear distinction between numerical examples and theoretical applications by moving the latter to Appendix G, labeled as "Additional Applications."
>
> ---
>
> > It isn't clear which results are classical, or almost classical with a few notational differences.
> For example, is Proposition 7 a new result? What is new in Section 4.2?
>
> Also:
>
> > The distinction between existing work and new contributions is not clear in 4.2. Can you write your results as a proposition?
>
> We have moved much of the technical detail to appendices, and tried to make clear what is classical and what is new. Everything in the main body of the document outside of background material in sections 2.1, 2.2, and the beginning of section 5 is new, to the best of our knowledge. We include some important classical results in the appendices to make the document as self-contained as possible, and have ensured these results are clearly cited in the text.
>
> In particular, Proposition 7 is new, as far as we are aware, though its proof is straightforward once the terminology and setting are established. To improve readability, we have moved the proof to an appendix and expanded the main text with additional exposition. Similarly, Section 4.2, which largely consisted of classical material, has been relocated to an appendix.

---

> ### Author Response · Authors · 2024-11-28
> **Questions and Suggestions**
>
> **Questions**
>
> > Fig 1 caption: Why does homotopy not give parts in kernel of $G$ that are not part of $F$ but somehow also correctly gives isolated point $(0,0)$?
>
> This is an excellent question. When executing the homotopy continuation method on $H$, we keep $x$ fixed while adjusting $y$ to stay in the kernel. For red dots corresponding to $x > 0$, $y$ must increase as $t$ increases to remain in the kernel; conversely, for red dots corresponding to $x < 0$, $y$ must decrease as $t$ increases. Furthermore, $H$ is chosen to have a vanishing time derivative at $(0,0)$ for all $t$, so the isolated point never moves. Thus even though the graph of $G$ is only a subset of the kernel of $H(x,y,1)$, we arrive at the correct "branch" for every $x$ at time $t=1$; the full path of the homotopy prevents any ambiguity.
>
> > What does "defined in pieces" mean?
>
> We have revised the sentence with this awkward phrase. The revised sentence is: "It is currently unknown whether every continuous real piecewise polynomial is ISD."
>
> > In corollary 8, "gr" is not defined.
>
> Thank you for pointing out this oversight, we have corrected it.
>
> > Why do we expect (7) to be small when $s < 1$?
>
> In the original manuscript, equation (7) referred to the "direction loss" we introduced to train the networks. We have completely rewritten this section, and hope it is clearer now. The direction loss will be small for $s < 1$ if the velocity vector $\dot{y}_j$ is pointing in the correct direction $y - y_j$.
>
> > Above Thm 16 you say "exactly compute" but in remark earlier you mentioned that this will still have discretization error due to ODE solve. Should this be changed?
>
> Thank you for pointing this out. In the revision we have been more careful in our choice of the words "compute" versus "represent". It is more accurate to say that SANNs can represent any bounded semialgebraic functions; but since there is approximation error introduced when solving the ODE (c.f. the Remark in Section 3.3), we should not claim to compute them exactly. In the revised manuscript we reserve the word "compute" to mean operations that are carried out exactly by a computer (up to the limit of floating-point arithmetic). The terminology has been updated throughout the manuscript.
>
>
> > Why does Theorem 14 only succeed with "probability 1"?
>
> We have replaced this phrase with the more accurate "...the family of homotopies in equation (11) is capable of representing any continuous bounded semialgebraic function $F$ for almost every choice of $a$; more precisely every $a$ except possibly on a semialgebraic set of dimension less than $n$. This small complication comes from the use of the Transversality theorem in Appendix D.1."
>
> **Suggestions**
>
> > Define acronym SANNS in main text.
>
> Thank you for pointing this out, we have corrected it.
>
> > Clarity of Definition 1 can be improved. Perhaps the free lattice should be defined?
>
> We have removed the work "free" here, since it is superfluous with the phrase that follows: "...generated by the polynomials $\mathbb{R}[x_1, \dots, x_m]$".
>
> > $C^k(D, R^n)$ is not defined (I'm assuming this is just k times differentiable functions on $R^n$?)
>
> It is indeed $k$-times differentiable functions from $D$ to $\mathbb{R}^n$. We have added this clarification to the text.
>
> > Does Section 4.1 really belong in Section 2?
>
> Section 4.1 contained a minor variation on a classical theorem in the field of homotopy continuation methods. We have moved it to an appendix.
>
> > In your summary of Section 4, perhaps include the subsection numbers to help navigate the relationships between the proofs.
>
> Thank you for the suggestion. Much of the technical content of Section 4 has been moved to Appendix D and E. We hope it is easier to follow the main ideas in the new revised section 4.
>
> ---
>
> We thank the reviewer once again for their thoughtful and constructive feedback. Their suggestions have helped us improve the clarity, organization, and overall quality of the manuscript, and we hope the revisions address their concerns.

---

### Official Review · Reviewer_MHwZ · 2024-10-31

**Soundness:** 3
**Presentation:** 3
**Contribution:** 3
**Rating:** 8
**Confidence:** 4

**Summary:**

This paper introduces a new class of neural network models, SANNs. The model is built on the basis of semialgebraic theory. The fead-forward network is defined using ODE and polynomial homotopy continuation method is used to ‘universality’ of the expressiveness of the neural networks. The paper focuses on theoretical aspects of the neural networks and contains abundant theoretical claims.

**Strengths:**

- Building a neural network based on semialgebraic geometry is very refreshing and definitely new class of ML models.
- Using (the idea of neural)ODE as a computational graph of SANN also makes a quite sense assuming polynomial homology continuation method is used.
- Visualization of the homotopy continuation method is really helpful to foster the readers’ understanding.
- Appendix covers exhuastive theoretical contents of the paper. Some applications are also discussed.

**Weaknesses:**

**Theoretical aspect:** In general, the paper assumes readers to be very familiar with semialgebraic geometry, which is (at least at this point in time) not a main stream of ML communities. I would presume the paper would be very hard to follow especially for those who are not familiar with this topic. I strongly encourage the authors to revise the paper so that the contents of the paper could be more accessible to even those unfamiliar with this topic. The followings are some concerns I found:
- The introduction of the paper is very hard to understand. Authors’ claims in the first paragraph of the introduction ‘’Semi algebraic functions include anything computable using …” and/or “Due to their ubiquity” would be unconvincing for those unfamiliar with this topic. It would be very helpful if the authors include references and/or brief explanation on the representative “classical numerical algorithms” that compute semialgebraic functions.
- The definition of lattice should be mentioned before Definition 1 or referring to Appendix A -- I would say that at least in ML-community a lattice typically means $ \mathbb{Z}^{d}$.
- ‘ker(f)’ in Proposition 7 should be also introduced. I would also say, the first sentence of the proof might be a bit rough -- It would be better to mention the continuity of $f$. Need to clarify the definition of $C^1$, since $\max ( 0, -q(x) )$ is generally not differentiable.
- The proof of Proposition 10 is hard to understand. Especially, ‘another ISD-function’ at line 311 is vague. Tarsi-Seidenberg theorem should be also stated somewhere in Appendix.
- What the authors mean by Training is somewhat unclear. When SANNs are trained, what parameters of SANNs are going to be trained? Does that mean the coefficients $a$ of $f_{k}(x)$ at line 154 are the trainable parameters?
- The claim in the lines 42-43 ‘neural networks capable of exactly computing any bounded semialgebraic function’ sounds vague since it is not clear what the authors mean by ‘exactly compute’. Does it mean, the proposed neural network can solve any root finding problems for semialgebraic functions?

**Experimental aspect:**
- The application that benefits from the proposed network is not clear and it is hard to evaluate numerical advantages of the method. Key advantages of the networks in numerical experiments are stated as that the computational complexity is low and the evaluation time is fixed when using (non-adaptive) ODE solver, either of which are not evaluated in experiments or important classes of applications. While the paper rather has a technical sound and supports the theoretical validity of those two aspects, having numerical experiments that support authors’ claim would add stronger experimental support of the method's advantage.
- I managed to find in line 332 ‘In our experiments, we found it insufficient to train using only the accuracy…’, but I cannot find results of any experiments.
- The example applications in Appendix G should be discussed in the main text.

**Minor:**
- ‘compute’ feels abused frequently. In some sentences, ‘represent’ instead of ‘compute’ sounds more convincing for me.
- L.20 or 21, “is able execute…”


**Overall comment (and meddling suggestion):**

The theoretical results of this paper are very intriguing and I believe this work would shed a light on the new usage of semialgebraic geometry in ML domain. On the other hand, I cannot overlook the major drawbacks that the paper lacks experiments which support theoretical finding and does not clarify numerical advantages against existing methods, as well as the lack of the readability of the paper. Therefore, I will give the paper the score ‘6’. However, I would be very comfortable to raise the score once those issues are addressed satisfactorily.

This may be none of my business, but I also cannot refuse wondering if this paper might also have more appropriate venue to be submitted given the abundance of theoretical contents. It might be very interesting if the authors could look for venues with more theoretical sound, such as (computational) mathematics journal. While the work of this paper definitely falls down into some category of machine learning domain, I also feel the paper would make a greater impact in the different communities;)

**Questions:**

- Why should semialgebraic function F not be simply modelled by ‘standard’ MLPs/NNs? In other words, why should F be encoded to the kernel of a piecewise polynomial?
- Do the authors have a (rough) estimation and/or observation about the computational complexity of fead forward and training of SANNs?
- What would be a concrete example and task in PDE simulation to which SANNs are expected to perform better than existing methods? What would be the advantage of using SANNs in this case?

---

> ### Author Response · Authors · 2024-11-28
> **On Weaknesses**
>
> Reviewers comment: In general, the paper assumes readers to be very familiar with semialgebraic geometry, which is (at least at this point in time) not a main stream of ML communities. I would presume the paper would be very hard to follow especially for those who are not familiar with this topic. I strongly encourage the authors to revise the paper so that the contents of the paper could be more accessible to even those unfamiliar with this topic.
>
> Answer: Thank you for your valuable feedback. We have revised the introduction where semialgebraic functions
> are discussed and  explained the terminology used in the theory of semialgebraic functions
> in  terms that are accessible for readers (emphasizing polynomials and iteration of the min and max functions).
>
>
> Reviewers comment: The introduction of the paper is very hard to understand. Authors’ claims in the first paragraph of the introduction ‘’Semi algebraic functions include anything computable using …” and/or “Due to their ubiquity” would be unconvincing for those unfamiliar with this topic. It would be very helpful if the authors include references and/or brief explanation on the representative “classical numerical algorithms” that compute semialgebraic functions.
>
> Answer:  We have added examples and references to applications semialgebraic functions in numerical linear algebra, such as solving linear systems, Schur complements, and extraction of real eigenvalues as well as in Finite Element approximations and perturbation theory of partial differential equations.
>
> Reviewers comment: The definition of lattice should be mentioned before Definition 1 or referring to Appendix A -- I would say that at least in ML-community a lattice typically means $\mathbb{Z}^d$.
>
> Answer:  We have emphasized that we use the term lattice as a family of functions that is closed
> in min and max operations and pointed out that this term has in this paper a  different use than
> the one used in physics and in several models used by the ML-community.
>
> Reviewers comment: ‘ker(f)’ in Proposition 7 should be also introduced. I would also say, the first sentence of the proof might be a bit rough - It would be better to mention the continuity of $f$. Need to clarify the definition of $C^1$, since $\max(0,-q(x))$ is generally not differentiable.
>
> Answer:  We have added the definition of the kernel notation, ‘ker(f)’, as the zero set of function f.
> We have also pointed out in the proof of Theorem 8 (Now in Appendix C) that the squared ReLU function
> $q\to (\max(0,-q))^2$ is in continuously differentiable. We have added brackets in the formula he avoid possible confusions with the notations.
>
>
> Reviewers comment: The proof of Proposition 10 is hard to understand. Especially, ‘another ISD-function’ at line 311 is vague. Tarsi-Seidenberg theorem should be also stated somewhere in Appendix.
>
>
> Answer: We have added a formulation of the Tarski--Seidenberg theorem in Appendix E9 (There are several equivalent formulations, so it is indeed a valuabel idea to present the formulation which we use). Also, we have add in the proof of Proposition 47 (now in Appendix E9), that the the ‘another ISD-function’ is the function $(z_j,\dot{z}_j)\to  z_j +\frac{1}{N} \dot{z}_j.$
>
> Reviewers comment: What the authors mean by Training is somewhat unclear. When SANNs are trained, what parameters of SANNs are going to be trained? Does that mean the coefficients $a$ of $f_k(x)$ at line 154 are the trainable parameters?
>
>
> Answer: We have added after the definition of function $f_k(x)$ in the lines  157-158 that the coefficients
>  $a_{k,i,j,\alpha_1,\dots,\alpha_m}\ $ are used as the parameters which are optimized in training process.
>
> Reviewers comment: The claim in the lines 42-43 ‘neural networks capable of exactly computing any bounded semialgebraic function’ sounds vague since it is not clear what the authors mean by ‘exactly compute’. Does it mean, the proposed neural network can solve any root finding problems for semialgebraic functions?
>
>
> Also:
>
> ‘compute’ feels abused frequently. In some sentences, ‘represent’ instead of ‘compute’ sounds more convincing for me.
>
>
> Answer: Thank you for the feedback. Indeed, we should have used the word "represent" rather than "compute." In the revision we have been more careful in our choice of these two words. It is more accurate to say that SANNs can represent any bounded semialgebraic functions; but since there is approximation error introduced when solving the ODE (c.f. the Remark in Section 3.3), we should not claim to compute them exactly. In the revised manuscript we reserve the word "compute" to mean either: (i) operations that are able to be carried out exactly by a computer (up to the limit of floating-point arithmetic), or (ii) the actual output of a network implemented on a computer. The terminology has been updated throughout the manuscript.

---

> > ### Author Response · Authors · 2024-11-28
> > **Experimental aspects**
> >
> > Reviewers comment: While the paper rather has a technical sound and supports the theoretical validity of those two aspects, having numerical experiments that support authors’ claim would add stronger experimental support of the method's advantage.
> >
> >
> > Answer: The revised manuscript contains numerical experiments in Section 5 and Appendix F.
> > In Section 5 we show that SANNs can solve linear systems, such that $Xy=g$. A SANN trained with standard techniques performs comparably to a feed-forward ReLU network on this task. This result highlights the feasibility of training SANNs. For the second demonstration, we apply SANNs in Electrical Impedance Tomography  problem for a electrical resistor network, where the voltage-to-current response map at some of the nodes is given and the task is to recover the resistivities  in all edges in the network. This problem is a discrete version of an inverse problems encountered in medical imaging.
> >
> > Reviewers comment: I managed to find in line 332 ‘In our experiments, we found it insufficient to train using only the accuracy…’, but I cannot find results of any experiments.
> >
> > Answer:  We apologize for the confusion in the original text. The revised manuscript contains numerical experiments in Section 5 and Appendix F.
> >
> > Reviewers overall comment and meddling suggestion: The theoretical results of this paper are very intriguing and I believe this work would shed a light on the new usage of semialgebraic geometry in ML domain. On the other hand, I cannot overlook the major drawbacks that the paper lacks experiments which support theoretical finding and does not clarify numerical advantages against existing methods, as well as the lack of the readability of the paper. Therefore, I will give the paper the score ‘6’. However, I would be very comfortable to raise the score once those issues are addressed satisfactorily.
> >
> > This may be none of my business, but I also cannot refuse wondering if this paper might also have more appropriate venue to be submitted given the abundance of theoretical contents. It might be very interesting if the authors could look for venues with more theoretical sound, such as (computational) mathematics journal. While the work of this paper definitely falls down into some category of machine learning domain, I also feel the paper would make a greater impact in the different communities ;)
> >
> >
> > Answer:  Thank you for your comment. The paper is of theoretical nature but we hope that as
> > semialgebraic functions are actively studied mathematical topic, our paper could link
> > the techniques of modern algebraic geometry to the machine learning research and
> > inspire the ML community to apply new types of mathematical techniques. In this way
> > the collaboration of ML community and mathematicians can put a huge body mathematical research in use of practical applications.

---

> ### Author Response · Authors · 2024-11-28
> **Questions**
>
> Reviewers comment:
>     Why should semialgebraic function F not be simply modelled by ‘standard’ MLPs/NNs? In other words, why should F be encoded to the kernel of a piecewise polynomial?
>
>
> Answer: The semialgebraic functions can precisely represent rational functions of several variables
> and also, the solutions of minimization problems for rational functions with constrains. Thus, many rigorous algorithms
> in linear algebra and in theory of partial differential operators (in particular, in inverse problems and parameter determination) can be written in terms of semialgebraic functions.
> However, one can ask, why it is beneficial to consider a family of neural networks that can
> implement also a rigorous algorithm instead of just using the rigorous algorithm developed
> using mathematical analysis. One answer to the rigorous algorithms do often work poorly with noisy data. Moreover, there are benefits to that the analytical algorithm can be precisely represented using the neural networks, and not only approximated (e.g. by Relu-based neural networks): When
>  the design of the architecture of the network is based on an analytic algorithm and
> this rigorous algorithm is thus included in the family of the considered neural networks, the trained
> neural network produces at least as good solution to the problem as the analytic algorithm.
> Thus the training produces a rigorously
> justified neural network which the properties (convergence, stabiilty, effect of errors) can be analyzed by studying the analytical algorithm and noting that the properties of the (optimally) trained algorithm is at least as good as the rigorous algorithm. Thus using SANNs instead of e.g Relu-networks can make the resulting algorithm not only explainable, but also rigorously analyzable.
>  SANNs solve the problem with noisy data better than the known analytical algorithm.
>
>
>
> Reviewers comment:
>     Do the authors have a (rough) estimation and/or observation about the computational complexity of feed forward and training of SANNs?
>
>
> Answer:  We have not yet theoretically analyzed the computational complexity. We plan to consider this
> in the future by using the corresponding results for operator recurrent neural networks. Numerical comparison of SANNs and feed forward networks are given in Section 5 and Appendix F.
>
> Reviewers comment:
>         What would be a concrete example and task in PDE simulation to which SANNs are expected to perform better than existing methods? What would be the advantage of using SANNs in this case?
>
>
>  Answer:  In Appendix F we consider an inverse problem by using a FEM approximation of and
>  elliptic PDE
>  $$
> \nabla \cdot (\sigma \nabla u) = 0,
> $$
> that models the electric voltage $u(x)$ is a body having the unknown conductivity function $\sigma(x)$. The task is to determine the function $\sigma(x)$ from boundary measurements
> of $u$ and its normal derivative.
> This inverse problem is encountered in the medical imaging and is called the Electrical Impedance Tomography. We demonstrate how SANNs can be used to solve a discretized version of this problem.
>
> Thank you very much for your valuable comments. Those were very important for us to improve the paper.

---

> ### Comment · Reviewer_MHwZ · 2024-12-03
>
> I thank the authors for their tremendous efforts to improve the paper. I found the presentation of the paper significantly improved and easier to understand, which made the paper very accessible to those new to this topic. The paper's main contribution is establishing a theoretical foundation for semialgebraic neural networks (SANNs.) With a similar spirit to MLP, the authors show that SANNs are universal approximators of fundamental classes of algebraic functions. Notably, SANNs can approximate any bounded semialgebraic functions which are possibly discontinuous, a new aspect distinct from conventional MLPs. In the rebuttal period, the authors also provide the results of several additional experiments. While the experiments sound rather elementary, the choice of the experiments sounds valid as the "POC" of the proposed concepts. Most of the experiments are reproducible (from the nature of simplicity of the systems) and potential applications are also introduced (in the appendix.) With the abundant theoretical results and the conceptually essential experiments, I believe this paper will be an excellent addition to the program. Hence, I will raise my score to 8. Thanks again, the authors, for conducting this wonderful work — I enjoyed reading the paper.

---

### Official Review · Reviewer_fqV6 · 2024-11-01

**Soundness:** 3
**Presentation:** 2
**Contribution:** 2
**Rating:** 5
**Confidence:** 3

**Summary:**

This paper proposes the use of polynomial homotopy continuation methods to design a novel neural network achitecture. Homotopy continuation methods consider a function $H$ which continuously deforms a target function $G_0$ until it reaches a target function $G$. This method uses neural networks with RELU activations to compute the vector field of an ODE corresponding to this process of continuous deformation.

The authors show formally that this method can compute all bounded semialgebraic functions and that their method can be extended to discontinuous functions.

Finally, a general class of optimization problems is specified, and the authors give example applications of their method.

**Strengths:**

1. The theoretical contributions of this paper seem strong. Capacity to compute all bounded semialgebraic functions (and extension to discontinuous functions) seem to be solid theoretical guarantees.
2. The proposed method seems, in principal, easy to implement and computationally efficient.
3. The method seems, in theory, to be applicable to a large range of optimization problems.

**Weaknesses:**

1. The subject matter presented in this paper is quite difficult, and I believe most ML researchers are probably quite unfamiliar with these mathematics. That being said, I found the exposition to be quite hard to read. Here are some more precise points/considerations I believe would improve the readability
    * Given this paper is presented to a machine learning (ML) conference, there should be more emphasis on the underlying learning problem. It might be interesting to motivate your architecture through a regression framework, for instance.
    *  You include many proofs in the main text and do not give much intuition on your mathematical results. I believe it would improve readability to put proofs in the appendix and spend more time developing intuition in the main text. You could for instance add more examples throughout the text such as the one presented in "Example 6".
    * I believe the algorithmic description of the architecture should be included in the main paper as it explains clearly the forward pass of your algorithm. It might be interesting to include

2. The paper offers no experimental results on the proposed architecture. Given the main contribution of this paper is a novel architecture, empirical validation demonstrating the trainability of the model is crucial. I believe adding simple experiments with synthetic data would increase the value of the paper. You could, for instance, define a simple regression problem with synthetic data generated by a semialgebraic function, and show your model can exactly recover the correct mapping.

3. "We give a few remarks on training SANNs, and leave a more thorough investigation for future work." Although you do define a loss objective to train the model, you provide no empirical or theoretical justification that this loss objective works.  As with point 2, I believe it would strengthen the paper to give a thorough investigation (either theoretical or empirical) of the training procedure. For notes on doing this empirically see point 2. This could also be done theoretically by analyzing the training dynamics for example.

4. I find the structure of the paper to be confusing. I will list my comments about paper structure below:
    * There is no discussion or conclusion: it would be nice to add a conclusion section in which you discuss limitations and address future directions of research.
    * The section which presents example applications does not contain any actual examples, only the general class of problems. It might help readability to use one of the example applications given in the appendix as a motivating example.
    * Most of the paper seems to be an introduction to semialgebraic geometry and homotopy continuation methods (section 2 and first half of section 4). Maybe it would help readability to give a less detailed/rigorous introduction to the field and focus on explaining how the architecture works/why this method is useful.

**Questions:**

* "To our knowledge, we present the first neural networks capable of computing arbitrary bounded semialgebraic functions on high-dimensional data." What advantage does such an architecture have compared to SoTA neural networks which are already widely used and easy to train?
* "In our experiments..." what experiments are you referring to? Would it be possible to include the discussed experiments in the final manuscript? If not please remove any references to experimental results
* You provide theoretical guarantees (Theorems 14 to 16) that your model can represent any bounded semialgebraic function. Do you have any intuition on learnability? Could your model learn a function exactly given a finite amount of samples?
* Can you give examples of how representing discontinuous functions exactly might be useful in practice?

---

> ### Author Response · Authors · 2024-11-27
> **Weakness 1 & 2**
>
> We thank the reviewer for carefully examining the paper and providing valuable criticism. Their suggestions have guided our revisions, and the manuscript is much improved as a result. In particular, we have focused on improving the clarity of the exposition, particularly in Section 3 where the SANN architecture is defined, and in Section 4 which outline the representation theorems. We have also added new numerical experiments, although the main contribution of the work remains the representation theorems. We hope these changes adequately address the reviewer's concerns and sufficiently improve the manuscript for acceptance.
>
> Please see the general comment summarizing the changes we made for this revision. We address further points raised by the reviewer below.
>
> **Weakness 1**
>
> > The subject matter presented in this paper is quite difficult, and I believe most ML researchers are probably quite unfamiliar with these mathematics. That being said, I found the exposition to be quite hard to read.
>
> Thank you for this feedback. We have moved some technical details to the appendices and expanded the exposition, particularly in Sections 3 and 4, to improve accessibility for a broader audience.
>
> > Given this paper is presented to a machine learning (ML) conference, there should be more emphasis on the underlying learning problem. It might be interesting to motivate your architecture through a regression framework, for instance.
>
> Our initial motivation was to design networks capable of executing classical numerical algorithms, which naturally led us to investigate semialgebraic functions. In Section 5 of the revised manuscript, we present an example of a classical algorithm implemented by a SANN: the Jacobi iteration for solving linear systems. We hope this example highlights both the motivation behind our work and the potential unlocked by the expressive power of SANNs. However, we acknowledge that further research on new training methods is necessary to fully harness this expressive potential.
>
> > You include many proofs in the main text and do not give much intuition on your mathematical results. I believe it would improve readability to put proofs in the appendix and spend more time developing intuition in the main text.
>
> We have relocated some of the more technical results to the appendices and expanded the exposition, which has noticeably improved the flow of the main manuscript.
>
> > I believe the algorithmic description of the architecture should be included in the main paper as it explains clearly the forward pass of your algorithm. It might be interesting to include.
>
> Thank you for this suggestion, we have incorporated the algorithmic description in Section 3.
>
> **Weakness 2**
>
> > The paper offers no experimental results on the proposed architecture. Given the main contribution of this paper is a novel architecture, empirical validation demonstrating the trainability of the model is crucial. I believe adding simple experiments with synthetic data would increase the value of the paper.
>
> We have included two new numerical examples: solving linear systems in Section 5 and a non-linear inverse problem in Appendix F. These examples demonstrate the feasibility of training SANNs using standard deep learning techniques, showcasing their potential for practical applications. However, we do not claim any numerical advantages at this stage. The main content of the paper is still the representation theorems. We hope the manuscript can be evaluated on these theoretical contributions.

---

> ### Author Response · Authors · 2024-11-27
> **Weakness 3 & 4**
>
> **Weakness 3**
> > Although you do define a loss objective to train the model, you provide no empirical or theoretical justification that this loss objective works. As with point 2, I believe it would strengthen the paper to give a thorough investigation (either theoretical or empirical) of the training procedure.
>
> We agree this is important. At this stage, we have numerical evidence the loss function works in the numerical results of Section 5 and Appendix F. We acknowledge the importance of further theoretical and empirical investigation into training techniques, which we outline as a priority in the new 'Discussion and future work' section.
>
> **Weakness 4**
>
> > There is no discussion or conclusion: it would be nice to add a conclusion section in which you discuss limitations and address future directions of research.
>
> Thank you for this suggestion. We have included two new sections: "Discussion and future work" and "Conclusion".
>
> > The section which presents example applications does not contain any actual examples, only the general class of problems. It might help readability to use one of the example applications given in the appendix as a motivating example.
>
> We have separated the "numerical examples" in section 5 and Appendix F from the "additional applications" in Appendix G.
>
> > Most of the paper seems to be an introduction to semialgebraic geometry and homotopy continuation methods (section 2 and first half of section 4). Maybe it would help readability to give a less detailed/rigorous introduction to the field and focus on explaining how the architecture works/why this method is useful.
>
> We have moved some of this content to appendices to give more room for expanded explanations.

---

> ### Author Response · Authors · 2024-11-27
> **Questions**
>
> > "To our knowledge, we present the first neural networks capable of computing arbitrary bounded semialgebraic functions on high-dimensional data." What advantage does such an architecture have compared to SoTA neural networks which are already widely used and easy to train?
>
> In the new Section 5, we give an example of a task performed by a SANN that is not possible for other architectures. We believe that being able to represent all bounded semialgebraic functions gives SANNs good inductive bias for tackling difficult problems, since many classical numerical algorithms are semialgebraic. However, we do not yet claim to have demonstrated numerical advantages when training from data, and acknowledge that future work is needed in this regard.
>
> ---
>
> > "In our experiments..." what experiments are you referring to? Would it be possible to include the discussed experiments in the final manuscript? If not please remove any references to experimental results.
>
> We apologize for the confusion caused by the original manuscript and have now included numerical results in this revision.
>
> ---
>
> > You provide theoretical guarantees (Theorems 14 to 16) that your model can represent any bounded semialgebraic function. Do you have any intuition on learnability? Could your model learn a function exactly given a finite amount of samples?
>
> This is an excellent question with intriguing connections to recent advances in algebraic geometry. Just as any $d$ points in $\mathbb{R}^2$ uniquely determine a degree-($d-1$) polynomial, a certain number of points in $\mathbb{R}^n$ can uniquely determine (up to certain conditions) an algebraic curve of a particular genus and degree. This classical problem has a rich history and has only recently been resolved (see [this article](https://www.quantamagazine.org/old-problem-about-algebraic-curves-falls-to-young-mathematicians-20220825/) contains an accessible explanation).
>
> Since the graphs of semialgebraic functions are composed of finitely many algebraic surfaces, these advances may provide insights into the complexity of a SANN required to exactly interpolate a given number of points. Developing this connection could yield a theoretical framework for understanding the learnability of semialgebraic functions by SANNs. The next step would then involve devising practical training techniques to leverage this theoretical foundation.
>
> While we lack the time and space to explore this fully in the current work, we view this as an exciting direction for future research and look forward to addressing it in greater depth.
>
> ---
>
> > Can you give examples of how representing discontinuous functions exactly might be useful in practice?
>
> Discontinuous functions frequently arise in scientific applications. While it is possible to approximate discontinuous functions in a suitable sense (e.g., in the $L^2$ norm) using continuous function classes such as polynomials or piecewise linear splines, the choice between continuous and discontinuous function classes has significant mathematical and computational implications.
>
> For a concrete example, the Finite Element Method (FEM) effectively approximates discontinuous solutions to hyperbolic systems of PDEs in a suitable sense. However, practical limitations of FEM have spurred interest in Discontinuous Galerkin (DG) methods, which offer greater flexibility for certain problems. In this analogy, classical neural networks are akin to FEM, while SANNs parallel DG methods, whose expressivity enables novel techniques and research directions beyond the capabilities of traditional architectures.
>
> We believe that demonstrating SANNs' ability to represent every bounded semialgebraic function (including discontinuous ones) ensures the theoretical results are both rigorous and self-contained, laying a strong foundation for future theoretical and practical advancements.
>
> Finally, while not all applications require discontinuous solutions, SANNs can easily be adapted to compute only continuous semialgebraic functions. For instance, ensuring the matrix $M$ in line 3 of Algorithm 1 is never singular guarantees continuity. We discuss one such approach in the new "Discussion and Future Research" section, but other modifications can be tailored to the specific problem at hand.
>
> ---
>
> We are sincerely grateful for the reviewer's detailed and constructive feedback, which has significantly enhanced the clarity and quality of our work.

---

> > ### Comment · Reviewer_fqV6 · 2024-12-03
> >
> > Thank you for taking the time to write this detailed response. After reading your comments as well as the other reviews. In light of the added experiments, and the improvements to the paper structure, I have chosen to increase my score to 5.
> >
> > Although the revised manuscript is much better than the first submission, I believe the additional experiments are not sufficient as they do not reflect the theoretical advantages you describe (i.e. capacity to compute any semialgebraic function). I personally would have hoped for an experiment justifying that the theoretical advantages of your method translate to gradient based training.

---

### Official Review · Reviewer_LTgJ · 2024-11-02

**Soundness:** 3
**Presentation:** 3
**Contribution:** 3
**Rating:** 8
**Confidence:** 3

**Summary:**

Authors proposed an architecture that is capable to approximate an arbitrary semialgebraic function. The main ingredients making the construction possible are two facts (i) it is possible to efficiently generate functions from ISD by neural networks with ReLU activations, (ii) any closed semialgebraic set (defining semialgebraic function) is a kernel of some ISD. Authors exploit these facts by combining a neural network with a homotopy method that deforms a simple semialgebraic set into a target semialgebraic set. The role of homotopy is to replace rootfinder that can be also used to find kernel of ISD. The resulting architecture resembles neural ODE, and requires adjoint for training. Authors demonstrated several theoretical results showing that with the proposed architecture it is possible to approximate continuous and discontinuous semialgebraic function.

**Strengths:**

The article introduces an interesting original idea that can be potentially used for many problems laying on the intersection of scientific computing and machine learning. Authors provide many details, examples and clarifications that help the reader with little background in semialgebraic approximation to better understand theory authors develop. Theoretical results seem to indicate that the proposed class of models represents a rather general set of functions. I also find particularly stimulating a discussion of relation between SANNs and deep equilibrium models solved with numerical continuation.

**Weaknesses:**

The article is theoretical, so the weak side is, naturally, a discussion of practical matters: computational complexity, how networks should be trained, how it compares with different related models, and questions alike. I put some of these questions in the section below, but overall I do not find this is a significant disadvantage, given that the goal of authors is to provide a certain "universal approximation" result for novel architecture they propose.

**Questions:**

1. In the introduction authors made an analogy with deep equilibrium model and write (line 46) "Therefore, in principle, we can use a neural network combining polynomials and ReLU activations to learn G, then append a root-finding procedure such as Newton’s method to compute $F (x) = \text{root}(G(x, \cdot)) = y$ in a manner similar to Bai et al. (2019)." Given that it seems that deep equilibrium models are already capable of computing arbitrary bounded semialgebraic function, and the claim that authors present first such method (line 59, "To our knowledge, we present the first neural networks capable of computing arbitrary bounded semialgebraic functions on high-dimensional data") is misleading. Can the authors please clarify this?
2. In line 270 and equation (2), (3) that jointly define SANN there is an operation defined as $M^{-1}b$ if $M$ is invertible and $0$ otherwise. This operation is not convenient from the numerical perspective, since it requires the inversion of potentially large matrices, besides it is not cheap to test numerically that $M$ is invertible. Is it possible to generate $M^{-1}$ right away from $ISD_{\text{net}}$ in place of $M$? Is it important that $M$ can be not invertible? Can the authors please clarify this part?
3. In line 332 authors write "In our experiments, we found it insufficient to train using only the accuracy of the final output $\left\|y^{\star} − y(1)\right\|$." No numerical experiments can be found in the article. Can the authors please describe numerical experiments they tried and show the results?
4. As I understand, training procedures include differentiation through the ODE solver. For that one needs to solve the adjoint ODE equation, and this is going to incur additional numerical costs. In a similar way, a deep equilibrium model defines an adjoint equation to obtain a derivative. Given that it is not clear what advantages the proposed method has over the deep equilibrium model (as discussed in the introduction) in terms of computations required. I kindly ask authors to discuss this issue.
5. In G2 authors provide interesting applications of SANNs. The idea, as I understand it, is to apply SANN to the space of equivalent feed-forward ReLU networks (which form a semialgebraic set). It is not entirely clear to me how this set can be constructed and manipulated with SANN. Can the authors provide more details, preferably, with some simple examples? It seems to me that this set is hard to work with, because practically indistinguishable networks can have weights arbitrarily distant in the $L_2$ norm https://arxiv.org/abs/1806.08459. It is also not clear to me how precisely SANNs can be used in the context of Lipschitz networks and for sparsifications.

---

> ### Author Response · Authors · 2024-11-28
> **Questions**
>
> Thank you for your valuable feedback. We hope that the paper provides
> information on new mathematical techniques for the ML community and have
> improved the introduction to make the paper more accessible.
>
> Reviewer's comment: In the introduction, the authors made an analogy with the deep equilibrium model and wrote (line 46)
>
> "Therefore, in principle, we can use a neural network combining polynomials and ReLU activations to learn $G$, then append a root-finding procedure such as Newton's method to compute $F(x) = \text{root}(G(x, \cdot)) = y$ in a manner similar to Bai et al. (2019).'"
>
> Given that it seems that deep equilibrium models are already capable of computing arbitrary bounded semialgebraic functions, the claim that the authors present the first such method (line 59, ``To our knowledge, we present the first neural networks capable of computing arbitrary bounded semialgebraic functions on high-dimensional data'') is misleading. Can the authors please clarify this?
> \end{reviewer}
>
> Answer: We have added references to  deep equilibrium models.
> SANNs are based on homotopy methods and thus they avoid problems related to local minimas.
> Thus, SANNs can produce in theory all points of the graphs of semialgebraic functions.
>
> Reviewer's comment: In line 270 and equations (2), (3) that jointly define SANN, there is an operation defined as $M^{-1}b$ if $M$ is invertible and 0 otherwise. This operation is not convenient from the numerical perspective, since it requires the inversion of potentially large matrices. Besides, it is not cheap to test numerically that $M$ is invertible. Is it possible to generate $M^{-1}$ right away from $ISD_{\text{net}}$ in place of $M$? Is it important that $M$ can be not invertible? Can the authors please clarify this part?
> \end{reviewer}
>
> Answer:  In the clamp operation that SANNs use, the possibly infinite values of $M^{-1}b$ are truncated.
> In numerical implementation, we can approxiamate the computation of  $M^{-1}b$ by using
> regularization techniques, which are equivalent to replacing the smallest singular values
> of $M$ by small parameter $\epsilon>0.$
>
>
> Reviewer's comment: In line 332, the authors write
> "In our experiments, we found it insufficient to train using only the accuracy of the final output $|y^* - y(1)|$."
> No numerical experiments can be found in the article. Can the authors please describe numerical experiments they tried and show the results?
>
>
> Answer: We have written the loss function used in training in an explicit way.
> The revised manuscript contains numerical experiments in Section 5 and Appendix F.
>
> Reviewer's comment: As I understand, training procedures include differentiation through the ODE solver. For that, one needs to solve the adjoint ODE equation, and this is going to incur additional numerical costs. In a similar way, a deep equilibrium model defines an adjoint equation to obtain a derivative. Given that it is not clear what advantages the proposed method has over the deep equilibrium model (as discussed in the introduction) in terms of computations required, I kindly ask the authors to discuss this issue.
>
>
> Answer: We have modified the definition of SANNs by replacing the continuous time ODE by a finite difference scheme. We hope that this clarifies the definition. We thank the reviewer on suggestions related to deep equilibrium models, and will work on the further comparison of these methods in the future.
>
>
> Reviewer's comment:  In Section G2, the authors provide interesting applications of SANNs. The idea, as I understand it, is to apply SANN to the space of equivalent feed-forward ReLU networks (which form a semialgebraic set). It is not entirely clear to me how this set can be constructed and manipulated with SANN. Can the authors provide more details, preferably with some simple examples? It seems to me that this set is hard to work with because practically indistinguishable networks can have weights arbitrarily distant in the $L_2$ norm (\href{https://arxiv.org/abs/1806.08459}{arXiv:1806.08459}). It is also not clear to me how precisely SANNs can be used in the context of Lipschitz networks and for sparsifications.
>
>
> Answer: Thank you for the valuable reference, we have added a citation to it and discussed the equivalent neural networks. We have also extended the discussion on how SANNs can be used to solve optimization problems.
>
> Thank you very much for your comments and suggestions. Those were very valuable to us in improving the paper.

---

### Author Response · Authors · 2024-11-27
**General reply**

We sincerely thank the reviewers for their thoughtful comments and valuable suggestions. We greatly appreciate the time and effort they invested in reviewing our work. We have carefully revised the paper and feel that it has significantly improved as a result of their feedback.

This general comment provides clarifications on issues raised by multiple reviewers and summarizes the main changes made in this revision. Specific responses to individual reviewer comments will be provided separately.

**Clarity.**
Our original definition of SANNs was difficult to understand. To address this, we have thoroughly revised Section 3 (where the architecture is defined) and included a new architecture diagram to better illustrate the relationships between its various components. While the design is inspired by tools from real algebraic geometry---particularly homotopy continuation methods for root-finding---we hope that we have now explained the architecture of SANNs in an accessible and conceptually clear way. With these revisions, we aim to make the explanation accessible to a broader audience, including those without previous background in real algebraic geometry, to easily understand and implement SANNs.

To further enhance readability, we have expanded the exposition in the main article and relocated some of the more technical content to the appendices.

**Implementation and numerical experiments.** Our original submission alluded to numerical experiments but did not include specific details or results. In this revision, we present numerical results for two experiments to illustrate the potential of SANNs.

As a first demonstration, in Section 5 we show that SANNs can solve linear systems: given a matrix $X$ and a vector $g$ as inputs, the network produces a vector $y$ such that $Xy=g$. A SANN trained with standard techniques performs comparably to a feed-forward ReLU network on this task, highlighting the feasibility of training SANNs. However, future work should focus on developing novel training techniques to fully leverage their expressive power.

To further showcase the possibilities, we manually configured the parameters of a SANN to replicate the classical Jacobi iteration for solving linear systems. This setup enables the SANN is able to solve the system to machine precision—something a standard feed-forward network, constrained to compute only piecewise-linear functions, cannot achieve.

For the second demonstration, we selected the electrical network EIT (Electrical Impedance Tomography) problem—a nonlinear inverse problem. Given an electrical network and the boundary voltage-to-current response map (i.e., the Dirichlet-to-Neumann map), the task is to recover the conductances of all edges in the network. This problem has broad applications, including in medical imaging, where it has enabled novel imaging modalities. Like the first example, a trained SANN performs comparably to a feed-forward ReLU network in this setting. This demonstration is in Appendix F.

**Conclusion and future work.** We consider the SANN architecture presented in this paper to be the most general form of a new family of neural networks designed to compute semialgebraic functions. In practice, this architecture can be specialized in various ways to improve performance on specific tasks. This is analogous to convolutional neural networks (CNNs), which are a subset of standard feed-forward neural networks. While CNNs have far fewer parameters and cannot compute arbitrary piecewise-linear functions, they excel at tasks involving translational symmetry (e.g., image classification). Similarly, in future work, we aim to identify and develop specialized variants of the SANN architecture that compute subsets of semialgebraic functions tailored to particular tasks. We discuss one such modification in the new "Discussion and Future Work" section (Section 6).

Efficient training of SANNs remains an open challenge. It is currently feasible to train SANNs using established techniques such as backpropagation or the adjoint sensitivity method. However, the unique structure of the SANN architecture offers opportunities for novel training strategies. New approaches are needed to take full advantage of the expressive power of SANNs.

---

### Meta-Review · Area_Chair_3zyB · 2024-12-19

**Metareview:**

The manuscript proposes a neural network architecture based on ideas of homotopy continuation method for roots finding. While there have been some concerns by the reviewers on actual practical performance based on gradient training, the reviewers find the methodology interesting. After carefully reading the revised manuscript and the discussions, the meta-reviewer tends to recommend acceptance of the paper in order to encourage novel neural network architecture designs.

**Additional Comments On Reviewer Discussion:**

The discussion was thorough.

---

### Decision · Program_Chairs · 2025-01-22

Accept (Poster)